# Wireless, battery-free, multifunctional integrated bioelectronics for respiratory pathogens monitoring and severity evaluation

Hu Li[1,2,11], Huarui Gong[3,4,11], Tsz Hung Wong[1,11], Jingkun Zhou[1,5,11], Yuqiong Wang[2,11], Long Lin[6], Ying Dou[4], Huiling Jia[1,5], Xingcan Huang[1], Zhan Gao[1], Rui Shi[1], Ya Huang[1,5], Zhenlin Chen[1], Wooyoung PARK[1], Ji Yu Li[1,5], Hongwei Chu[1], Shengxin Jia[1], Han Wu[2], Mengge Wu[1], Yiming Liu[1], Dengfeng Li[1], Jian Li[1], Guoqiang Xu[1], Tianrui Chang[2], Binbin Zhang[1,5], Yuyu Gao[1], Jingyou Su[1], Hao Bai[7], Jie Hu[7], Chun Ki Yiu[1,5], Chenjie Xu[1], Wenchuang Hu[7] ✉, Jiandong Huang[3,4,8,9] ✉, Lingqian Chang[2,10] ✉ & Xinge Yu[1,5] ✉

The rapid diagnosis of respiratory virus infection through breath and blow remains challenging. Here we develop a wireless, battery-free, multifunctional pathogenic infection diagnosis system (PIDS) for diagnosing SARS-CoV-2 infection and symptom severity by blow and breath within 110 s and 350 s, respectively. The accuracies reach to 100% and 92% for evaluating the infection and symptom severity of 42 participants, respectively. PIDS realizes simultaneous gaseous sample collection, biomarker identification, abnormal physical signs recording and machine learning analysis. We transform PIDS into other miniaturized wearable or portable electronic platforms that may widen the diagnostic modes at home, outdoors and public places. Collectively, we demonstrate a general-purpose technology for rapidly diagnosing respiratory pathogenic infection by breath and blow, alleviating the technical bottleneck of saliva and nasopharyngeal secretions. PIDS may serve as a complementary diagnostic tool for other point-of-care techniques and guide the symptomatic treatment of viral infections.

Severe acute respiratory syndrome coronavirus 2 (SARS-CoV-2) has triggered an unprecedented contagious respiratory illness that spreads primarily through breath and blow (e.g., speaking, cough, sneeze), resulting in more than 6.9 million deaths worldwide (https://covid19.who.int/)[1–6]. It is widely believed that the genome mutations and antigenic evolution will increase the risk of vaccination failure and reinfection[7,8], life-threatening to the given populations with comorbidities[9,10]. The rapid diagnosis of virus infection and symptom severity provides scientific basis for timely establishing epidemic prevention policies and appropriately regulating medical resources in public health emergencies at an early stage. Up to now, the miniaturized wearable/portable bioelectronic technologies for rapidly diagnosing SARS-CoV-2 infection and symptom severity by breath and blow remain underdeveloped and challenging[2]. The unsolved

technical barriers exist in the rapid collection and onsite identification of the exhaled SARS-CoV-2 by the platform itself, multi-parameter monitoring and in-depth analysis, miniaturized system integration and non-invasive severity assessment after virus infection.

The pioneering studies related to breath stay in the simulation and the laboratory antigen screening using commercial proteins and synthetic RNA fragments[11,12], 2 h of visual lateral flow assay of nucleic acid signatures[12], narrow product selectivity[11–13] and energy-intensive system[11] that restrict applications to the low-resource settings[5,14]. Here we present developments (see Supplementary Tables 1 and 2, Supplementary Note 1) in bionic breath and blow sample collection, multifunctional sensing design, machine learning analysis and battery-free system integration that enables rapid multifunctional virus infection analysis by breath and blow within one minute without the need of sample pre-treatment. We carry forward the technique to practical human diagnosis beyond the laboratory assay. This technical progress in diagnosing respiratory pathogenic infection by breath and blow can overcome the technical bottleneck of saliva and nasopharyngeal secretions. It may simplify the screening process of virus infection and guide patients to receive symptomatic treatment and follow-up, and further contribute to lowering the morbidity of severe infections in the long-term coexistence with SARS-CoV-2.

## Results
### Design concept of diagnosing SARS-CoV-2 infection by breath and blow
The infected hosts produced viral aerosols on different sites of the respiratory tract and emitted SARS-CoV-2 into environment by breath and blow[5,6]. Meanwhile, the virus infection will elevate the respiration rate (RR) and exhaled breath temperature (EBT) because of the fever and inflammatory response that reflects the symptom severity (Fig. 1a)[15–20]. By monitoring and deeply analyzing the exhaled virus, abnormal RR and EBT with the machine learning, we can rapidly diagnose the SARS-CoV-2 infection and symptom severity by breath and blow without the need of saliva, nasopharyngeal secretions, and blood sampling[21–36].

The viral aerosol suspended in air after exhalation, which was naturally trappable and dissolvable in aqueous liquid via the air-liquid interface because of its aqueous core-shell structure[5]. The dissolved virus can be specifically captured by spike antibodies modified on the activated graphene as an immuno biosensor (IBS) (inset, Fig. 1a). Here the use of graphene for biosensing was due to the wide demonstrations in biochemical assays for its superior conductivity and large surface area[23,25,31,32]. In addition, graphene likewise showed a sensitive response behavior to the adsorption and desorption of aerosols and thus recorded the respiration characteristics as a respiration biosensor (RBS)[37,38]. The inbuilt temperature biosensor (TBS) in the NFC circuit was used to measure the EBT. The integration of IBS, RBS and TBS modules into the final wireless, battery-free multifunctional PIDS allowed to monitor the various signals of viral aerosols, RR and EBT (Fig. 1b). The IBS and RBS used the same polyimide (PI) substrate and served as the front-end biosensing modules. TBS served as the back-end biosensing module. The IBS module consisted of a bionic microchannel, spike antibody, molecular linker, graphene, interdigital electrodes, PI film and steel sheet from up to down. The RBS module was based on graphene, interdigital electrodes, PI film and steel sheet from up to down. The front-end and back-end modules were connected by a FPCB (flexible printed circuit board) connector (Fig. 1b).

Figure 1c showed the module diagram explaining the operational principle of the integrated PIDS. The transponder of the PIDS collected data and can deliver them toI/IEC 15693 standard enables the smartphone to wirelessly power and communicate the PIDS by an electromagnetic field while reading out data from the NFC chip. The analog-to-digital converter (ADC) converts the biosensing information to digital signals. The microcontroller unit (MCU) with a 2 MHz central processing unit (CPU) controls the sampling rate and packages the collected data into NFC Data Exchange Format (NDEF) which is accessible by smartphones or other types of radio frequency readers.

The PIDS has a strong evolution ability when it diagnoses the virus infection by breath and blow. It can be combined with the face mask as an inbuilt platform for active at-home self-diagnosis and passive public diagnosis by breath test with the smartphone touch and walking through the wireless biosafety doors, respectively (Fig. 1d, Supplementary Fig. 1, Supplementary Movie 1 and Movie 2). It can be also transformed into other independent platforms as the handheld breathalyzer or personalized electronic patch to identify the virus infection by blow test (Fig. 1d). Figure 1e–l showed pictures of the wireless, battery-free multifunctional PIDS for breath analysis and its evolved platforms for blow test, including the independent PIDS (Fig. 1e), inbuilt PIDS in a face mask (Fig. 1f), breathalyzer (Fig. 1g), wristband (Fig. 1h), watch (Fig. 1i), mini detection kit (Fig. 1j), necklace (Fig. 1k) and electronic patch (Fig. 1l). These products are packaged with silicone elastomer to form a water-proof encapsulation and protect the electric elements from failure when they suffer from the moist breath and blow. The package can also mechanically isolate the modules to achieve a gentle interface with the skin. The detailed preparation process was shown in Supplementary Figs. 2–4.

### Bionic microchannel for rapid collection of breath and blow samples
The design of the microchannel for breath and blow samples collection was inspired by the protein distribution of coronavirus (Fig. 2a, b). The distributions of inlets and outlets mimic the shape of spike proteins around the envelope, while the air channels mimic the insertion of membrane proteins in the envelope. The directions of the air channels were marked using red ink infill (Fig. 2c). The PDMS with a wrinkled and hydrophobic surface (Fig. 2d, e) contributed to keeping the air-liquid interface stable and protecting it from flowing away when shaking happens during walking around (Supplementary Fig. 5).

To realize the high efficiency in collecting aerosols, the inlet number, outlet length and width, as the key parameters, were simulated. The inlet number was studied from 2 to 13 that met the demands for enough aerosol entrance. More inlets admitted more aerosol inflows into the chamber. Correspondingly, the pressure at the air-liquid interface and airflow velocity at the outlet end increased with the inlet number from 1.7 Pa to 11.2 Pa and 0.1 m/s to 0.7 m/s, respectively (Fig. 2f, g and Supplementary Figs. 6 and 7). A higher pressure accelerates the dissolution of aerosols in PBS (phosphate buffer saline). However, the excessive pressure tends to blow the PBS away from the chamber (Supplementary Figs. 8–10). Nine inlets with an appropriate pressure (ca. 8 Pa) were chosen here as the optimal inlet number. The outlet length was studied from 1 mm to 11 mm (Fig. 2h, i and Supplementary Figs. 11 and 12). The pressure increased with the outlet length. While the airflow velocity decreased first when the outlet length was less than 7 mm, and then it reached the plateaus after 7 mm, where the pressure is about 8 Pa. Similar with the selection criteria of inlet number, the 7 mm was adopted as the optimal length parameter. The outlet width was studied from 0.3 mm to 3 mm (Fig. 2j, k, Supplementary Figs. 13 and 14). The pressure and airflow velocity decreased first with the outlet width and then kept stable when the width exceeded 1 mm, where the pressure is about 8 Pa. Accordingly, the 1 mm was adopted as the optimal width parameter.

Finally, we verified the capture capability of aerosols through the optimized microchannel using an atomizer (Fig. 2l and Supplementary Fig. 15). Moderate amounts of PBS were pre-injected into the reservoir chamber to build an air-liquid interface. We used a handheld atomizer filled with red ink to produce the aerosols that can be pressed into the air channel through the inlets and entered the chamber under the action of pressure difference arisen from the atomizer (or breath and blow)[39–41]. Parts of the aerosols skimmed

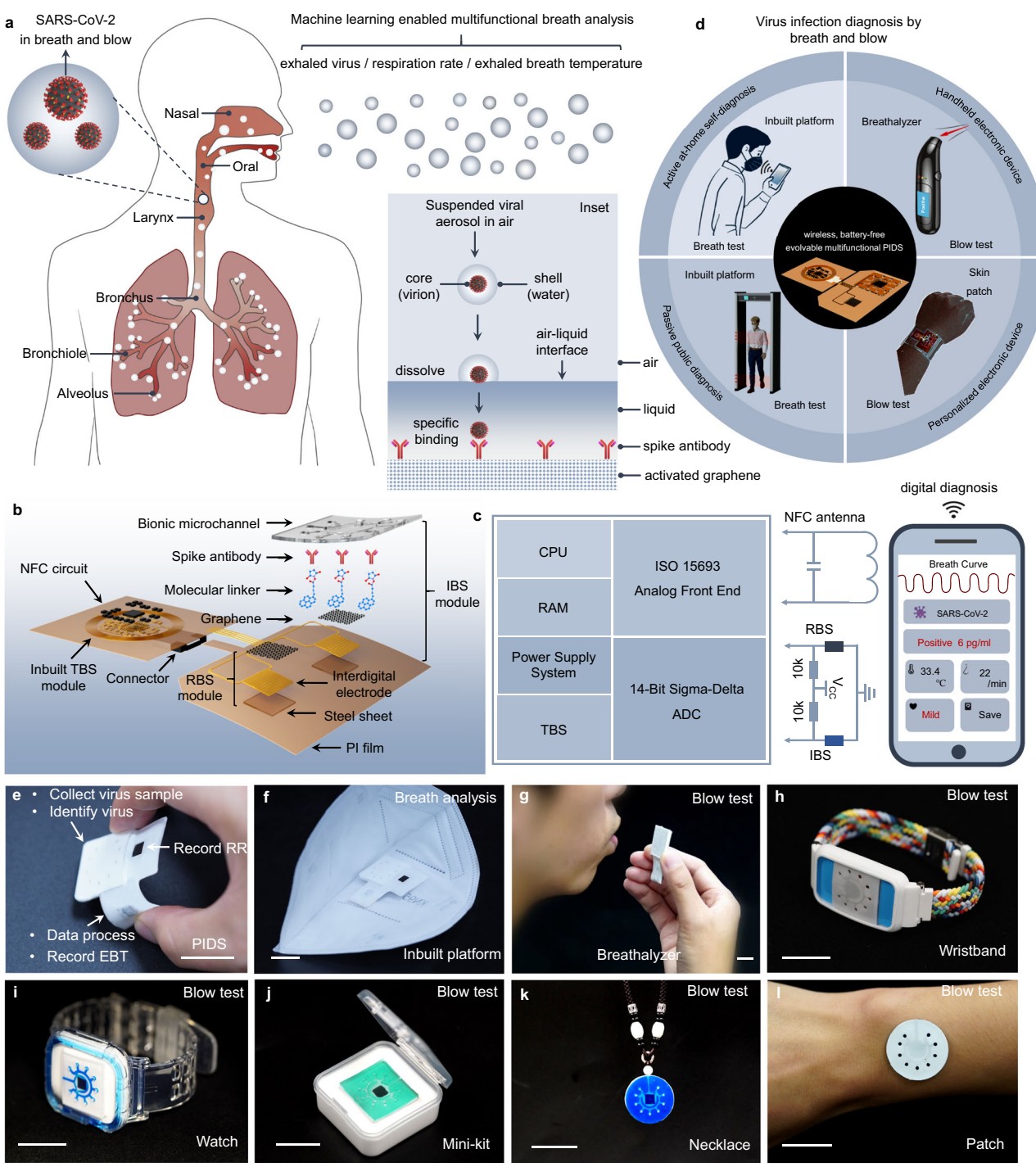

**Fig. 1 | Overview of the wireless, battery-free, multifunctional, evolvable pathogenic infection diagnosis system (PIDS) for diagnosing virus infection by breath and blow. a** Diagram illustrating the generation and exhalation of the virus by breath and blow. The inset interprets the mechanism of collecting viral aerosols and binding the released virus by building an air-liquid interface. **b** Structure, components, and multiple modules of PIDS for monitoring exhaled virus and recording respiration rate (RR) by immune biosensor (IBS) module and exhaled breath temperature (EBT) by temperature biosensor (TBS) module. **c** Block diagram of PIDS consists of the central processing unit (CPU), analog-to-digital converter (ADC), random access memory (RAM), power supply, analog front end, temperature biosensor (TBS), respiration biosensor (TBS), immune biosensor (IBS) and near-field communication (NFC) antenna. The health information can be wirelessly read out using the NFC-enabled smartphone. **d** Outlook of virus infection diagnosis by breath and blow test using the evolved PIDS platforms. **e** Picture of the multifunctional PIDS platform. It was packaged with white PDMS. **f**–**l** The evolved PIDS platforms as miniaturized wearable, handheld, personalized electronics can be used in multiple diagnostic modes. **f** PIDS as an inbuilt platform in a face mask for breath analysis. **g** PIDS as a handheld breathalyzer for blow test. **h, i** PIDS as the wearable wristband and watch for blow test. **j** PIDS as a rapid virus detection kit for blow test. **k, l** PIDS as the personalized necklace and skin patch for blow test. Scale bar, 2 cm. The abbreviations are the same in other figures unless otherwise stated.

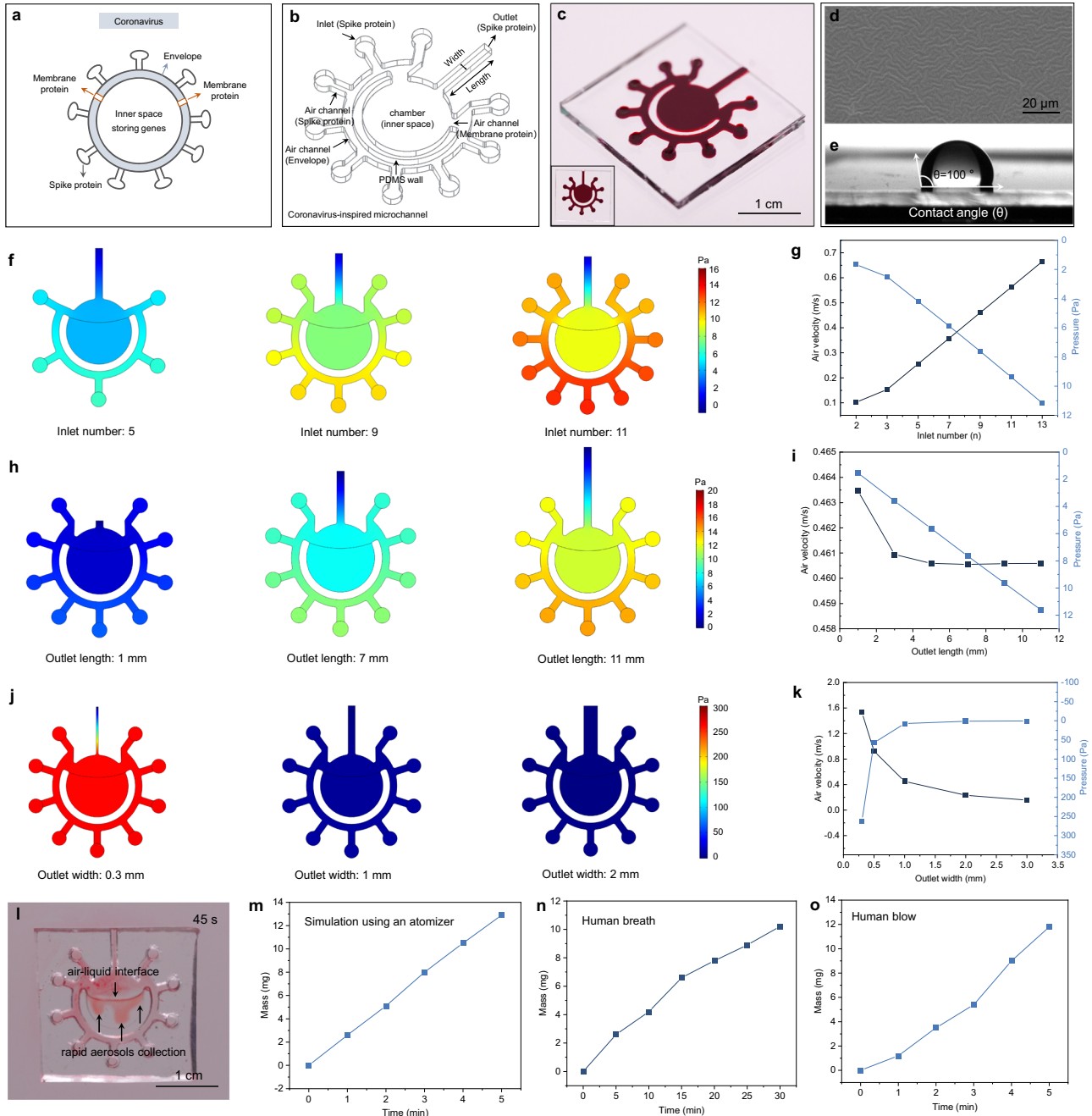

**Fig. 2 | Coronavirus-inspired microchannel and the rapid collection of breath sample and blow sample. a** Structure diagram of the coronavirus involving the envelope, spike protein, membrane protein, and inner space for storing genes. **b** Structure diagram of the coronavirus-inspired microchannel with the inlet, outlet, air channel and reservoir chamber. **c** Picture of a transparent PDMS microchannel marked using red ink. **d, e** Surface morphology and hydrophobic property of the microchannel, respectively. The contact angle (θ) of water on PDMS surface is 100°. Scale bar is 20 μm. **f, g** Pressure simulation and the correlation of the pressure and

air velocity with different inlet numbers from 2 to 13, respectively. **h, i** Pressure simulation and the correlation of pressure and air velocity with different outlet lengths from 1 mm to 11 mm, respectively. **j, k** Pressure simulation and the correlation of pressure and air velocity with different outlet widths from 0.3 mm to 3 mm, respectively. **l** Demonstration of aerosol collection using the optimized microchannel. **m** Collection speed of aerosols using an atomizer. **n, o** Collection speed of aerosols by human breath and blow, respectively. Source data are provided as a Source Data file.

over the interface and dissolved in PBS. The excess aerosols ran outside through the outlet. An observable amount of red aerosols were captured at the interface within 15 s (Supplementary Fig. 15). Plenty of red aerosols and diffusion occurred at 45 s via the air-liquid interface (Fig. 2l). The aerosols filled up the PBS within 1 min (Supplementary Fig. 15). The average collection speed of the simulated aerosols was about 2.6 mg/min. About 13 mg of aerosols were captured in the channel within 5 min (Fig. 2m). It also showed outstanding collection abilities of breath samples and blow samples in

human experiments (Fig. 2n, o). The average collection speeds of breath and blow are about 0.33 mg/min and 2.4 mg/min, respectively. The collection speed decreased with the distance between the atomizer and microchannel both in an enclosed space and an open space, which could be attributed to the reduced airflow velocity at a longer distance. For the blow test, the temperature difference between the exhaled aerosols and environment in the open space was more beneficial to collect the aerosols than the enclosed space (Supplementary Figs. 16 and 17).

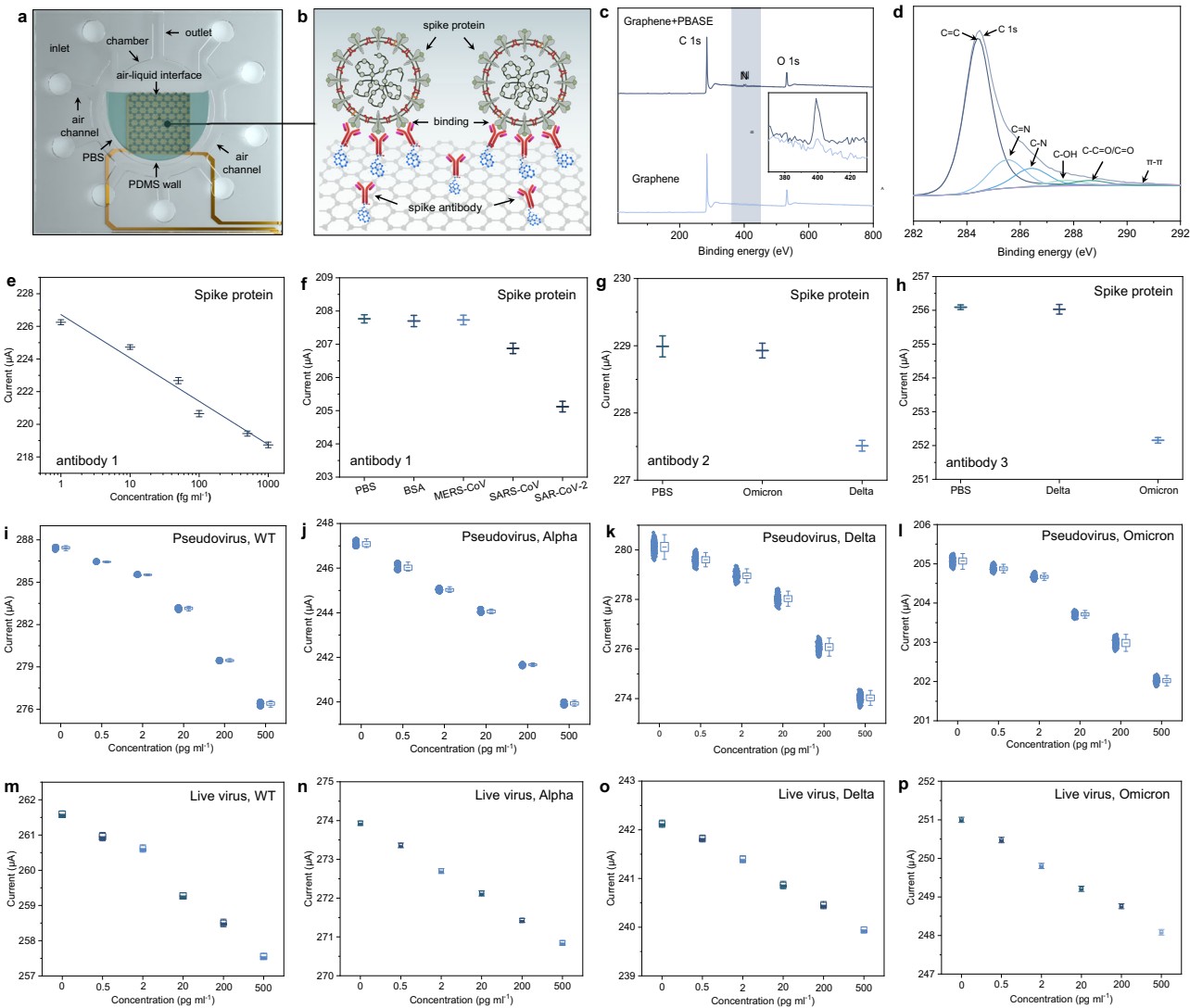

**Fig. 3 | Surface modification and detection performance of the PIDS for SARS-CoV-2 spike proteins, pseudo virus and live virus. a, b** Reaction principle of the SARS-CoV-2 with antibodies modified on graphene surface in PBS solution. **c** Elemental analysis of the graphene before and after PBASE modification using X-ray photoelectron spectroscopy. **d** C1 peaks of PBASE activated graphene from 282 eV to 292 eV, including C=C, C=N, C-N, C-OH, C-C=O/C=O and π-π bonding. **e** Dose-dependent current curve of PIDS to SARS-CoV-2 spike proteins. Graphene was modified with antibody 1 (spike S1 antibody, Cat: 40150-R007) that was reactive with wide type (WT), Alpha, Delta and Omicron. The antigen protein was spike S1 His Recombinant Protein (Cat: 40591-V08H). **f** Specific detection of PIDS to SARS-CoV-2. The used antigen proteins involved BSA, MERS-CoV, SARS-CoV and SARS-CoV-2. **g, h** Selective identification of Delta and Omicron by modifying antibody 2 (Cat:40592-MM57) and antibody 3 (Cat:40591-MM48) on graphene, respectively. **i–l** Detection ability of PIDS modified with antibody 1 to pseudo viruses including WT, Alpha, Delta and Omicron. The box ends represent the standard deviation (SD). The horizontal line in each box represents the mean value. The upper and lower whiskers refer to the range of non-outlier data values. Outliers were plotted as individual points. **m–p** Detection ability of PIDS modified with antibody 1 to the live viruses including WT, Alpha, Delta, and Omicron. The experiment was conducted in physical containment level 3 laboratory. The concentration range was from 500 fg ml⁻¹ to 500 pg ml⁻¹. The error bars in **e–h** and **m–p** correspond to the SD. The average current values and error bars were calculated using the recorded 1200 stable current values (*n* = 1200) of each test. Data are presented as mean values ± SD. Source data are provided as a Source Data file.

## Identification of spike proteins, pseudovirus and live virus of mutated variants

Figure 3a, b show the interaction principle of specific antigen-antibody binding of SARS-CoV-2 in PBS. The aerosols collected by the microchannel release viruses into PBS after dissolution through the air-liquid interface (Fig. 3a). The spike antibodies on the activated graphene specifically bind with spike proteins of SARS-CoV-2 (Fig. 3b), which lowers the surface resistance of graphene and converts the biochemical signal into the electrical signal via the back-end circuit. A molecular linker, 1-pyrenebutyric acid succinimidyl ester (PBASE), was used to activate the graphene surface and further combine with antibodies[32,42–44]. PBASE activates the graphene through π-stacking of the pyrene group onto its surface[32,43,44]. In addition, the succinimide

portion of PBASE facilitates the immobilization of antibodies via conjugation reactions between the amine groups of antibodies and the amine-reactive succinimide group of PBASE[42].

The activation increased the line roughness ($R_{ms}$) and surface roughness ($R_q$) of graphene from 16.9 nm to 47 nm (Supplementary Fig. 18a) and 43.9 nm to 106 nm, respectively (Supplementary Fig. 18b). XPS (X-ray photoelectron spectroscopy) reflects the element core levels of C 1s, O 1s and N 1s. An obvious N 1s peak can be observed and clearly enhanced after PBASE activation (Fig. 3c and right inset). EDX (Energy Dispersive X-ray) mapping also proves the characteristic nitrogen (N) element on the activated graphene (Supplementary Fig. 18c, d). Additionally, the six types of C 1s bonding peaks formed from 282 eV to 292 eV, including C=C, C=N, C-N, C-OH, C-C=O/C=O and

π-π conjugation, also verified the successful functionalization of PBASE on the graphene surface (Fig. 3d).

The spike antibodies further combine with PBASE on graphene (Supplementary Fig. 19) and are blocked with BSA (bovine serum albumin). Three spike antibodies can be separately modified for universally identify the variants of concerns (VOC) and selectively identify the VOC, respectively. The limit of detection (LOD) was first measured using SARS-CoV-2 spike S1 antibody (Cat: 40150-R007, hereinafter called the "antibody 1") that is reactive to the pandemic VOC including WT (wild type), Alpha, Delta and Omicron (Fig. 3e). The concentration range of the spike protein is from 0.5 fg/ml to 1 pg/ml that covers the minimum identifiable concentration. Therefore, the IBS module achieves the distinguishable LOD of 1 fg/ml (Supplementary Fig. 20a). The stable response current shows an approximate linearity with the protein concentrations (Fig. 3e), which can be used to roughly calculate the virus loads in the chamber using the smartphone. We further measured the specificity of antibody 1 modified IBS module using similar coronaviruses involving Middle East respiratory syndrome coronavirus (MERS-CoV) and severe acute respiratory syndrome coronavirus (SARS-CoV) (Fig. 3f and Supplementary Fig. 20b)[43]. IBS shows no response to the control groups (PBS and BSA) and MERS, but it generated a distinguishable current response to SARS-CoV and SARS-CoV-2. This result was also reported in previous work[25], which could be attributed to the highly protein identity (76–95%) of SARS-CoV and SARS-CoV-2. However, the homology was only 30–40% for MERS-CoV[45–47]. The identification ability to SARS-CoV was also declared by the antibody supplier (https://cn.sinobiological.com/antibodies/cov-spike-40150-r007). In practical applications, this special case for SASR-CoV will not interfere the specificity of IBS module to SARS-CoV-2 due to the nonexistence of SARS-CoV in COVID-19 pandemic.

Except the universal identification ability, the IBS module also exhibits good selectivity by identifying Delta and Omicron by modifying SARS-CoV-2 spike neutralizing antibody (Cat. 40592-MM57, hereinafter called the "antibody 2") and SARS-CoV-2 spike neutralizing antibody (Cat. 40591-MM48, hereinafter called the "antibody 3"), respectively. The antibody 2 specifically bound to Delta spike protein but not to Omicron spike protein (Fig. 3g and Supplementary Fig. 20c). The antibody 3 specially bound to Omicron spike protein but not to Delta spike protein (Fig. 3h and Supplementary Fig. 20d). These results indicate that the IBS module enables the identification of the virus species by modifying corresponding antibodies. Additionally, when the concentration of spike protein was lower than the LOD, the response current curves overlapped with the baseline current. When the spike protein was not reactive with the antibody, the response current curves also overlapped with the baseline current (Supplementary Fig. 20a–d). These results indicated that the IBS module was stable without drift during the test. When the protein was reactive with the antibody, the current decreased with time at the high concentration (500 fg/ml in Supplementary Fig. 20c, d), which was because that the efficient dynamic binding between the antibody and protein was enhanced gradually with incubation time[48]. More spike proteins can bind with the antibodies over time.

Pseudoviruses of four pandemic VOCs were selected to verify the universality of IBS to detect SARS-CoV-2, including the wild type (WT), Alpha variant, Delta variant and Omicron variant (Fig. 3i–l and Supplementary Fig. 21). IBS module shows a LOD of 0.5 pg ml$^{-1}$ in the concentration range of 2 fg ml$^{-1}$ to 500 pg ml$^{-1}$ (Supplementary Fig. 21). Although the mutation degree increases from WT to Omicron[49,50], IBS can still rapidly identify all the variants within 50 s when the concentration is higher than the LOD due to the outstanding binding ability of antibody 1 to the spike protein (Fig. 3i–l and Supplementary Fig. 21). The detection capability also applied to the live virions (Fig. 3m–p and Supplementary Fig. 22a–d), IBS module responds to all the live virions with a distinguishable current reduction. Studies reported that a SARS-CoV-2 virion weighs about 1 fg[51], the concentration of 0.5 pg/ml in the

chamber with 40 μL of PBS can be equivalent to 20 virions. Considering that the number of virion-laden aerosols released from the patients per minute is in the range of 1000–100,000[5,6], thus the IBS module is capable of identifying the exhaled SARS-CoV-2.

## Human diagnosis by breath and blow tests

Afterwards, we applied the PIDS to the identification of positive and negative cases and recorded their respiration rate and exhaled breath temperature. We recruited 21 positive cases (percentage, 50%) and 21 negative cases (percentage, 50% = 38.1% +11.9%). Sixteen negative cases (percentage, 38.1%) have never been infected. Five negative cases (percentage, 11.9%) have been infected before and fully recovered health when they participated in the test (Fig. 4a). The ages of the positive cases are in the range of 20–59 (Fig. 4b). Most of the participants have been vaccinated with Sinovac, Comirnaty and their combinations (Fig. 4c, d). Two of the positive cases have not been vaccinated (Fig. 4c). The vaccination indicated that the vaccine cannot prevent the infection of SARS-CoV-2 variants. The rapid diagnosis of SARS-CoV-2 infection is necessary and important to reduce the morbidity of severe infection at an early stage. The other related information including the age, gender, RR, EBT, symptoms and medicine intake were summarized in Supplementary Tables 3 and 4 and Supplementary Fig. 23. When the positive cases breathe with the PIDS (Fig. 4e) or directly blow the IBS module of PIDS (Fig. 4f), the current decreased gradually with test time (Fig. 4e, f and Supplementary Figs. 24–26). The identification time by blow is much shorter than that by breath, which can be attributed to the higher aerosol collection efficiency of blow (Fig. 2n, o). In contrast, the current kept stable for the negative participants whether they conducted a breath test or a blow test (Fig. 4g, h and Supplementary Figs. 27–29). We first tested the negative participants and then statistically analyzed the current change ratios using the boxplot (Fig. 4i, j). The percentage change was calculated by the formula, $(I-I_0)/I_0 \times 100\%$, where I and $I_0$ were the measured average stable current and primary stable current, respectively. The current change ratios of negative cases just fluctuated within a narrow range and no drift occurred during the repeated tests, proving the stable performance of IBS under the breath and blow conditions. Then we tested the positive cases and compared them with the negative cases (Fig. 4i, j). According to the difference in change ranges, the percentage change of −0.2% (blue dotted line) was selected as the threshold value to divide and redefine the positive and negative cases. When the amount of the percentage change was greater or less than −0.2%, they were redefined as "positive" or "negative", respectively. It can thus be concluded that the PIDS can identify the positive cases within 1 min and 5 min by blow and breath, respectively (Fig. 4k). The diagnosis accuracy was calculated using the formula, $Accuracy = \frac{N_{accurate}}{N_{total}} \times 100\%$, where the $N_{accurate}$ is the accurate diagnosis number of participants. $N_{total}$ is the total number of participants. Participants could exhale more and more viruses with the test time. More and more viruses could be captured by the IBS module. Therefore, the diagnosis accuracy increased with the test time. The accuracies for blow tests at different times are 73.8% at 1 min, 92.9% at 2 min and 3 min, 100% at 4 min and 5 min, respectively (Fig. 4l). The accuracies for breath tests at different time are 64.3% at 5 min, 88.1% at 10 min, 95.2% at 15 min, 97.6% at 20 min, 100% at 25 min and 30 min, respectively. The overall accuracy of all blow tests for 5 min and all breath tests for 30 min are 91.9% and 90.9%, respectively (Fig. 4l). Figure 4m, n showed the morphology of captured viruses in the chamber by breath and blow, respectively. It further proved the feasibility of utilizing the IBS for capturing exhaled viruses. Figure 4o, p showed the pictures of one negative volunteer and one asymptomatic volunteer conducted self-diagnosis using the PIDS by breath. The PIDS rapidly and accurately diagnosed them including the RR, EBT, viral infection and symptom severity just by smartphone touch (Supplementary Movie 3 and Movie 4).

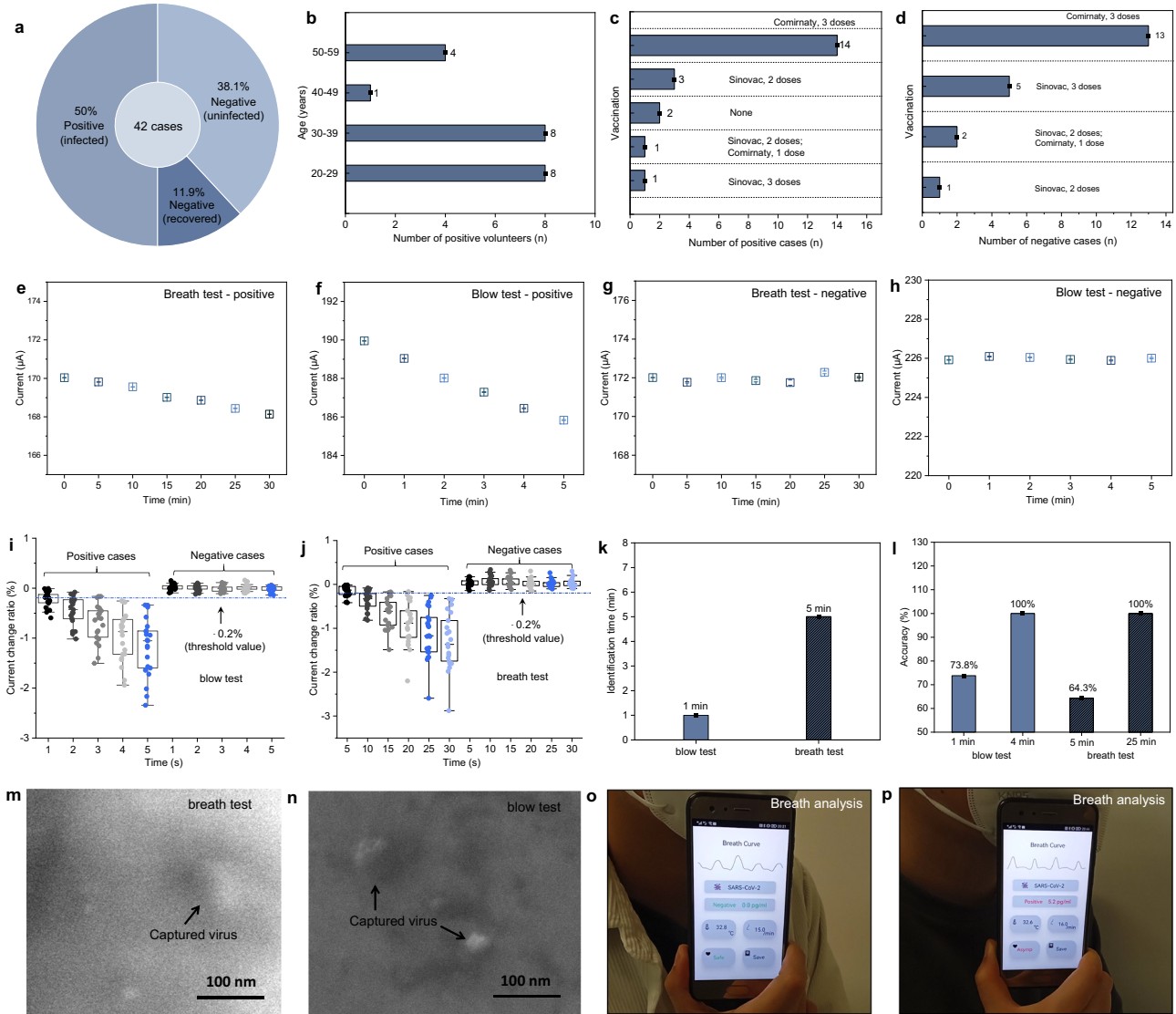

**Fig. 4 | Rapid and accurate identification of positive and negative cases by breath test and blow test. a** 42 volunteers were recruited in this experiment. **b** Age distribution of the positive cases. **c, d** Vaccination of positive and negative volunteers when they participated in the test, respectively. **e, f** Typical current change of the positive volunteers by breath test and blow test, respectively. The current values gradually decreased with the test time. **g, h** Typical current change of the negative volunteers by breath test and blow test, respectively. The current values kept stable during the tests. No obvious decrease or increase occurred. **i, j** Statistic analysis of the current values obtained by blow test and breath test of 42 volunteers, respectively. There are 21 data points for each test group. The box ends represent the 25th and 75th percentiles. The horizontal line in each box represents the median. The upper and lower whiskers refer to the range of non-outlier data values. Outliers were plotted as individual points. **k** The identification time of virus infection by blow test and breath test are 1 min and 5 min, respectively. **l** The accuracies of blow test are 73.8% at 1 min and 100% at 4 min for blow test, and 64.3% at 5 min and 100% at 25 min, respectively. **m, n** The virus captured in the microchannel by breath and blow, respectively. The black arrows indicated the viruses. Scale bar: 100 nm. **o, p** Rapid diagnosis of SARS-CoV-2 infection by breath test using the wireless and battery-free PIDS. The health information was digitally displayed on the smartphone. **o** Negative case. **p** Positive case. "Asymp" represents "asymptomatic". The error bars in **e**–**h** correspond to the standard deviation. The average current values and error bars were calculated using the recorded 1200 stable current values ($n = 1200$) of each test. Data are presented as mean values ± SD. Source data are provided as a Source Data file.

## Machine learning evaluation of infection and severity by breath analysis

Machine learning is an effective tool to automatically extract features from the time-domain data of positive/negative cases and precisely identify the infection and classify the symptom severity[3]. Figure 5a is the workflow showing the details of the evaluation of virus infection and symptom severity using PIDS with the assistance of machine learning. We collected and preprocessed the data acquired from the participants using the three modules (i.e., IBS, RBS and TBS) of PIDS by breath test. After feature extraction and selection, we established the machine learning model between the feature matrix and the labels,

and then iterated continuously. The machine learning model gradually approached the ground truth when increased the training data volume. After the model was established, the PIDS could further predict the infection and symptom severity. Figure 5b showed the working mechanism of RBS by adsorption or desorption of aerosols on graphene that increased or decreased the surface resistance, respectively[37,38]. When a constant voltage was applied on the RBS module, the output current decreased and increased periodically with the breath. It enabled RBS module to sensitively record the respiratory characteristics during the periodic exhalation and inhalation (Fig. 5c and Supplementary Fig. 30a). The collected signals were wirelessly

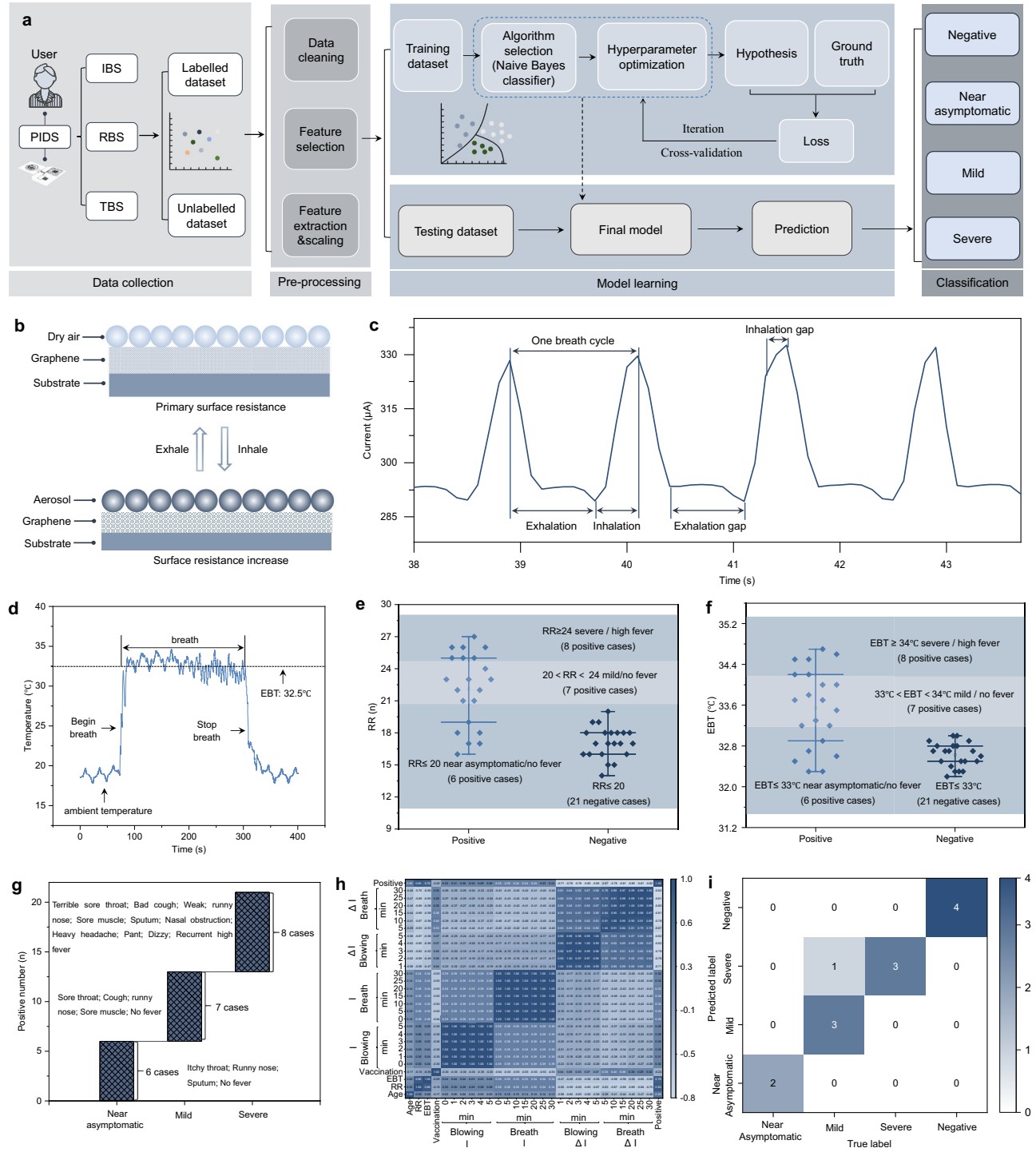

**Fig. 5 | Evaluation of SARS-CoV-2 infection and symptom severity by machine learning enabled multiparameter breath analysis. a** Workflow of the machine learning enabled diagnosis of virus infection and symptom severity. **b** Recording mechanism of RR by adsorption and desorption of aerosols on graphene. **c** A typical breath curve recorded by RBS module at a rapid rate of 45 min⁻¹. **d** A typical temperature curve recorded by the TBS module. The average value during breath was adopted as the EBT. **e**, **f** Statistical analysis of RR and EBT for 42 cases, respectively. **g** Bridge map was used to divide the symptom severity of positive cases into three categories according to the RR, EBT and related symptoms. **h** Correlation heatmap of 28 variables and positive/negative diagnosis. The RR, EBT and current change ratio (i.e., ΔI) showed a high correlation with positive diagnosis. The age, vaccination, and current value (i.e., I) showed a low correlation with positive diagnosis. **i** Confusion matrix to predict the positive/negative cases and symptom severity through the machine learning model. Source data are provided as a Source Data file.

transmitted to the mobile phone by NFC technique. The peak detection algorithm can be applied to identify the crest locations of each respiratory wave. By counting the number of peaks within a given time window, the respiratory rate can be derived and defined as the number of breaths taken per minute (Supplementary Figs. 31 and 32).

The breath rate, breath depth, exhalation gap and inhalation gap were all clearly recorded even at a fast respiration rate of 45 min⁻¹. Besides the rapid breath, the RBS module also held for recording normal breath and heavy breath (Supplementary Fig. 30b). Furthermore, the RBS showed universality to different individuals in

monitoring the respiration characteristics (Supplementary Fig. 30c). The breath curves of 42 cases are listed in Supplementary Figs. 33–40. For the negative cases, most of the current intensities (amplitude) and rhythms are similar and regular due to the unobstructed breath. Some irregular current intensities and rhythms occurred in the curves for the cases 5, 6, 17, 18 because of the rhinitis (nasal obstruction). This phenomenon is witnessed more frequently among positive cases with nasal obstructions (runny nose, cough). The normal respiration rates of healthy adults are between 12 min$^{-1}$ and 20 min$^{-1}$ [52].

Figure 5d presented the temperature response against the TBS module, as tested by beginning breath and stopping breath. When recording the ambient temperature, some low-frequency noise peaks (~0.04 Hz) appeared in the curve (Fig. 5d; Supplementary Fig. 41a). These peaks could be filtered out using the notch filter (Supplementary Fig. 41b). The average temperature during breath was adopted as the EBT value. The statistical analysis indicated that the RR and EBT of eight positive participants with high fever, terrible sore throat, bad cough, and related symptoms are higher than 24 min$^{-1}$ (RR ≥ 24) and 34 °C (EBT ≥ 34 °C), respectively (Fig. 5e, f). The RR and EBT of seven positive participants without fever but with cough, sore throat and related symptoms were in the range of 20–24 min$^{-1}$ (20 < RR < 24) and 33–34 °C (33 °C < EBT < 34 °C), respectively. The twenty-one negative participants and six positive participants just with itchy throat, runny nose and sputum showed similar values for RR and EBT, which are less than 20 min$^{-1}$ (RR ≤ 20) and 33 °C (EBT ≤ 33 °C), respectively. Fever usually causes tachypnea, febrile convulsion and even death to children and the elderly, which makes it a predictive symptom related to mortality in the ICU (intensive care unit)[53]. Here, we divided the "severe" and "mild" based on the fever symptom. Further, we divided the "mild" and "asymptomatic" according to the secondary symptoms (e.g., sore throat, cough, sore muscle) (Fig. 5g). Then we analyzed the correlations between 28 variables and viral infection (i.e., positive) by calculating Pearson correlation coefficient (PCC) (Fig. 5h). The results indicated that RR, EBT and the amount of current change (ΔI) showed high correlation (PCC ≥ 0.7 or PCC ≤ −0.67) with the positive diagnosis. These variables can be effective indicators for identifying positive/negative cases and classifying the symptom severity. In contrast, age, vaccination, and current values had a low correlation (PCC < 0.4) with positive/negative diagnosis whether the participants conducted the breath test or blow test. Together with the correlation analysis, we use the established machine learning model to evaluate the virus infection and symptom severity in the confusion matrix (Fig. 5i and Supplementary Fig. 42). The accuracy was calculated by the formula, $Accuracy = \frac{Number\ of\ correct\ predictions}{Total\ number\ of\ predictions} \times 100\%$. We randomly selected 13 cases from the participants to predict their health status and the symptom severity. The model differentiated the positive and negative cases with a 100% accuracy rate and predicted their symptom severity with a 92% accuracy rate, proving that the PIDS assisted with the machine learning was qualified for the rapid and accurate non-invasive diagnosis of respiratory pathogenic infection and symptom severity. It may guide the personalized therapy after infection and lowering the mortality at an early stage.

## Discussion

Breath and blow are the major transmission routes of SARS-CoV-2 that provide the access priority for diagnosing respiratory infectious diseases[5,6,54] and alleviating the testing bottlenecks of point-of-care techniques relied on saliva, nasopharyngeal secretions, and blood[9–24]. We developed a wireless, battery-free, multifunctional PIDS platform for diagnosing SARS-CoV-2 infection and symptom severity by breath and blow. The PIDS platform integrated the bionic microchannel, immuno biosensor, respiration biosensor, temperature biosensor and battery-free system into a miniaturized all-in-one system. It can not only implement infection screening but also evaluate the symptom severity. PIDS with machine learning gave prompt diagnosis just by

smartphone touch, guiding patients to receive symptomatic treatment and follow-up. Compared with traditional swab sampling and amplification techniques[4], the breath and blow analysis in our work is easy to operate and no special training required. It dispensed with manual sampling, complicated pretreatment, expensive equipment, time-consuming transportation, frequent replacement of power supply, and resource-intensive laboratory assays[54].

We also demonstrated the versatile evolution of PIDS as the miniaturized wearable/handhold/personalized detection platforms that promote its alternative option and accessibility in multiple life scenarios. It provided more choices for noninvasively diagnosing and predicting the viral infection and symptom severity at an early stage. Considering the similar transmission routes, PIDS can be also extended to other respiratory infectious diseases (e.g., influenza) by replacing the antibodies (Supplementary Fig. 43), proving the general applicability of our technology in detecting and preventing virus infection at an early stage. In future efforts, the accuracy of the breath and blow analysis can be potentially improved by incorporating more bioinformation in breath and blow, for example, the volatile organic compounds[55]. Other measures involving expanding the training dataset and exploring more machine learning algorithms[56] may enhance the generalizability to a larger population.

To ensure the accuracy and long-term stability of PIDS during practical applications, some measures can be considered: (1) Pretest the device before the blow or breath diagnosis to confirm its validity; (2) Keep the chamber of IBS module in the liquid condition and store the device at a negative temperature (e.g., household refrigerator) to protect the antibody's activity and ensure the long-term stability for several months; (3) Increase the testing time to achieve the optimum accuracy. For example, 4 min for blow test; 25 min for breath test; (4) Replace new products regularly in case of unconscious damage or destruction. The last but not the least, the IBS won't be reusable after testing the positive cases to avoid the virus transmission caused by the contaminated device. While the IBS is still reusable if the tests showed negative results. Furthermore, the IBS can be still active by storing it at room temperature for three days (Supplementary Fig. 44) or in the refrigerator at −20 °C for several months.

## Methods

### Ethical statement

All participants provided informed consent when they took part in this research. All human experiments were performed in accordance with protocols approved by the Institutional Review Board of the University of Hong Kong/Hospital Authority Hong Kong West Cluster (UW 23-107). The authors confirm that the human research participants provided written informed consent for the publication of the image in Fig. 1g and Supplementary Table 2, and the publication of the Supplementary Movies.

### Fabrication of the interdigital electrodes

Polyimide (PI) film (size, 10 cm × 10 cm; thickness, 300 μm) was cleaned using acetone, ethanol, and deionized water, respectively. The appropriate amounts (1.5 ml) of poly-amic acid solution (12 wt% ± 0.5 wt%) were spin-coated onto the PI film at 3000 rpm for 30 s. Then the samples were heated on a hotplate at 250 °C for 30 min. Gold (200 nm) was deposited on PI film using the dual target sputtering system (QUORUM Q150TS). A positive photoresist (PR, AZ 4620, AZ Electronic Materials) was first spin-coated onto the gold at 3000 rpm for 30 s and baked at 100 °C for 5 min. Then, the sample was sent for UV exposure for 45 s with a cover of a photo mask using the lithography machine (URE-2000/35AL deep UV, IOE, CAS). Afterwards, the sample was developed in AZ 400 K developer solution for 90 s to remove the exposed photoresist. The bare gold was etched using gold etchant (I$_2$/KI solution, I$_2$: KI: water = 1:4:40). The remaining photoresist was washed off by acetone and deionized water. The prepared

samples were cut into proper sizes for standby applications. The dimension was shown in Supplementary Fig. 45a, b. The electrode width and gap are 150 μm and 50 μm, respectively. There are twenty electrodes in total for each interdigital electrode. Considering that the dissolution of viral aerosols was initiated from the air-liquid interface, the parallel design of interdigital microelectrodes helps to capture viruses on multiple pairs of microelectrodes and weaken the influence of the uneven diffusion and binding of virus on graphene electrodes.

## Fabrication of the IBS module and RBS module

The prepared interdigital electrodes were fixed on a heated hotplate at 140 °C. A polyethylene terephthalate (PET) mask was used to cover the blank area to protect them from spraying. Two square openings (0.5 cm × 0.5 cm) were made to expose the electrodes to graphene spraying. Then the graphene dispersions in water (500 μL, 1 mg/ml) were sprayed on the interdigital electrodes. Afterwards, the samples were immersed in deionized water for 12 h to remove the surfactant (sodium dodecyl benzene sulfonate, SDBS) and dried on a hotplate at 80 °C. Then two stainless steel sheets (1 cm × 1 cm × 1 mm) were fixed at the bottoms of IBS and RBS using A + B glue to protect them from external mechanical interference. PDMS (purchased from Dow Corning Corporation) with white silicone pigment (Smooth-On, Inc.) was poured onto the device to form a package layer (~500 μm), which can protect them from delamination and achieve the friendly contact with the skin in practical application. The square puncher (0.5 cm × 0.5 cm) was used to cut and remove the PDMS covered on graphene. The left and right graphene areas were used for IBS and RBS, respectively. Next, 20 μL of 1-pyrenebutyric acid N-hydroxysuccinimide ester (PBASE; Sigma-Aldrich) solution in methanol (5 mM) was drop-cast onto the graphene surface and incubated for 2 h at room temperature. Some extra methanol was drop-cast around the graphene to reduce the evaporation rate. Then the graphene was washed with methanol and deionized water, respectively. The spike antibody (10 μl, 250 μg/ml) of SARS-CoV-2 was drop-cast onto the graphene and incubated for 3 h at room temperature (25 °C). Finally, the graphene was rinsed with deionized water and then blocked with 2% BSA (Bovine Serum Albumin; Sigma-Aldrich) protein. The activated graphene was used for specific binding with SARS-CoV-2. The cleaning of SDBS together with the modification of PBASE, antibodies and BSA will lead to some differences in the baseline currents between different IBS modules. The baseline currents of IBS module in PBS varied between 160 μA and 350 μA. In our experiment, we adopted the current change ratio (not the absolute current value) to monitor the capture of virus, which can eliminate the effect of difference in sensing data baselines. The results in the following tests also proved this point that the difference in baseline currents showed no influence on the detection of viral infection (Fig. 3e-p; Supplementary Figs. 20–22, 24–29). The right bare graphene without any modification was used as the breath sensor. The IBS module was covered with a bionic PDMS microchannel using A + B glue (epoxy resin). Then the PBS (30 μL-40 μL) was injected into the chamber (Supplementary Fig. 46) to cover the activated graphene and protect the antibody activity. The samples were stored in the refrigerator at −20 °C for further usage. There is no observable evaporation and leakage occurred during the storage (Supplementary Fig. 47). The IBS module was not affected by the PBS volume and could keep a stable baseline current in different volumes of PBS (Supplementary Fig. 48). Additionally, the IBS module could keep stable baseline current and showed no current drift under the unknow breath and blow conditions (e.g., sudden breath and blow) (Supplementary Fig. 49).

## Fabrication of the bionic microchannel

The square and circular molds were prepared by 3D printing technology using UV-curable resins. The mold surfaces are sputtered with gold (100 nm) to ensure the curing ability of PDMS in the molds. PDMS (resin: curing agent = 10:1) with white pigment, blue pigment, or green pigment was poured into the molds and heated at 100 °C for 5 min. The transparency is adjustable by adding different amounts of pigments. The detailed dimension of the microchannel was shown in Supplementary Fig. 50.

## Assembly of the integrated PIDS

The PI film with front-end biosensors (IBS and RBS) was connected to the NFC circuit by the FPCB connector. Then the device was encapsulated with PDMS and cured at 80 °C for 10 min. The cured PDMS covered on IBS and RBS were removed using the square puncher. Then the immunosensor was modified using above-mentioned method. Finally, the bionic PDMS channel was used to cover the IBS with A + B glue.

## Fabrication of the NFC circuit

The NFC circuit was fabricated using flexible printed circuit processing techniques of copper on PI film. The circuit has an insulating treatment to protect it from an electrical short circuit. The electric elements are bonded and electrically connected by low-temperature solder paste. An NFC sensor transponder (RF430FRL152H, TI.) was used to collect data from sensors and transfer them to the user interface via the NFC antenna. By adopting the ISO/IEC 15693 standard, this NFC chip can be powered by the electromagnetic field and transfer data to the user interface when a smartphone is placed in the range. The signals generated on the sensors can be converted to digital signals by the onboard 14-bit sigma-delta analog-to-digital converter (ADC). Those digital signals are buffered in Random Access Memory (RAM) inside the NFC chip, waiting to be read. A 16-bit MSP430 low-power MCU (microcontroller unit) with a 2 MHz CPU (central processing unit) inside the NFC sensor transponder was adopted to control the sample rate and package the collected data into NFC Data Exchange Format (NDEF) so that a smartphone can easily read it even without installing the mobile application for this device. The resonance frequency of the NFC circuit is 13.56 MHz. It keeps stable at around 13.56 MHz at different bending angles (Supplementary Fig. 51). The detailed circuit design was shown in Supplementary Fig. 52.

## Measurement of the SARS-CoV-2 spike proteins, pseudovirus and live virus

The spike proteins and antibodies are purchased from Sino Biological, including SARS-CoV-2 spike protein (Sino Biological, Cat:40591-V08H), MERS-CoV spike protein (Sino Biological, Cat:40069-V08H), and SARS-CoV spike protein (Sino Biological, Cat: 40150-V08B1); Spike antibody 1 (Sino Biological; Cat: 40150-R007; Clone ID: 007; Dilution, 1:4), spike antibody 2 (Sino Biological; Cat:40592-MM57; Clone ID: 57; Dilution, 1:4) and spike antibody 3 (Sino Biological; Cat: 40591-MM48; Clone ID: 48; Dilution, 1:4). Influenza A H1N1 HA protein (Sino Biological, Cat: 11055-V08H) and influenza B HA protein (Sino Biological, Cat: 11053-V08H). Influenza A H1N1 HA antibody (Sino Biological; Cat: 11055-MM11; Clone ID: 8F3G7; Dilution, 1:4) and influenza B HA antibody (Sino Biological; Cat: 11053-R004; Clone ID: 004; Dilution, 1:4). All the spike proteins are diluted to different concentrations using PBS (1X). The primary antibody concentration is 1 mg/ml. The antibodies were diluted into 250 μg/ml when modify the graphene. The pseudovirions and live virions are diluted to different concentrations using DMEM (Dulbecco's Modified Eagle Medium). 30 μL of the solution at each concentration was added onto the sample surface and measured for 5 min. A constant voltage of 0.2 V was applied to the immunosensor using an electrochemical workstation (CHI660E). For the live virions, the experiment was carried out in the biosafety level 3 lab of the University of Hong Kong. The immunosensor was connected to the NFC circuit, the smartphone was used to read the information for 5 min for each concentration.

## Culture of the SARS-CoV-2 pseudovirus and live virus

**Pseudovirus.** Co-transfection of HEK293T cells (ATCC, Cat: CRL-3216) with psPAX2 (Addgene, 12260), pLenti-luciferase (Addgene, 17477), and plasmids containing the ancestral SARS-CoV-2 spike (eEnzyme, Cat: SCV2-PsV-001) protein, Alpha spike (B.1.1.7, reference sequence: EPI_ISL_601443, eEnzyme, Cat: SCV2-PsV-UK), Delta spike (AY.44, reference sequence: EPI_ISL_6416262, Codex BioSolutions, Cat: CB-97100-161), or Omicron spike (BA.1.15, reference sequence: EPI_ISL_6699757, Codex BioSolutions, Cat: CB-97100-167) with polyetherimide resulted in the production of pseudovirions. The cell culture medium at 40 h and 64 h post-transfection were extracted, centrifuged at 600 g for 10 min, filtered through a 0.45 μm membrane to remove cell debris, and then frozen at −80 °C for further use.

**Live virus.** SARS-CoV-2 WT, Alpha, Delta, and Omicron were isolated from specimens of confirmed COVID-19 patients in Hong Kong. All SARS-CoV-2 variants were cultured and titrated in VeroE6 (ATCC, Cat: CRL-1568)-TMPRSS2 (Sino Biological, Cat: HG13070-CH) cells by plaque assays and stored at −80 °C. All live virus experiments were performed following the approved standard operating procedures of the Biosafety Level 3 facility of the University of Hong Kong.

## Characterization of materials

The surface morphology of graphene was observed using the scanning electron microscope (SEM, HITACHI, SU8020) and atomic force microscope (AFM, Asylum Research, MFP-3D). The nitrogen element was analyzed using energy dispersive X-ray (EDX; IXRF SYSTEMS, Model 550i) mapping at 20 kV and 10 μA. The chemical binding components were characterized using X-ray photoelectron spectroscopy (XPS, Thermo Scientific K-Alpha) with the Al K Alpha source gun at 12 kV and 6 mA. The hydrophobic property of PDMS was measured using the contact angle meter (Dataphysics OCA20).

## Characterization of SARS-CoV-2 morphology

The breath and blow samples were collected in the chamber of PIDS within 5 min and 1 min, respectively. Then the 4% paraformaldehyde (PFA) solution was added into the PBS solution to inactivate the SARS-CoV-2 viruses. 1 μL of the PBS was added on the p-type silicon wafer coated with poly-L-lysine without gold sputtering. After the natural evaporation of PBS, the sample was washed with deionized water to remove the salts in PBS. Then naturally evaporate the sample for further characterization by scanning electron microscope (SEM).

## Human experiments of breath and blow tests

We did the human tests for 4 months. 21 positive volunteers and 21 negative volunteers have been recruited. 10 males and 11 females are involved in the positive tests. 10 males and 11 females are involved for the negative tests. The gender of participants was determined based on self-report. The age and other information were listed in Supplementary Tables 3 and 4. The gender was not considered in the study design because the virus transmission and exhalation have no certain relationship with gender after viral infection. All the PIDS stored in the refrigerator at −20 °C can work normally during the period of human test. The test of positive participants was carried out when they were self-isolated at home at 25 °C controlled by the household air conditioner. All the positive participants have completed the food intake and medicine intake for about 2 h. Two methods were used to test the identification ability of PIDS. The first method was to fix the PIDS on the flat side of the KN95 face mask. Then the participants wore the face mask and recorded the health information using the smartphone every 5 min. The second method was to directly blow the inlets of IBS module and record the health information using the smartphone every 1 min. The distance between mouth and the microchannel was 1 cm. The blow number was 20 times per minute. The time was 3 s for each blow cycle. The negative participants were tested in the lab with the same operation steps. All the negative participants have completed the food intake for about 2 h. Additionally, the ability of PIDS to differentiate the biomarker from normal human activities (e.g., food intake) was also tested during food intake (Supplementary Fig. 53). All participants have provided informed consent when they took part in this experiment. 200 HK dollars per hour are paid as the compensation for all participants.

## Statistics and reproducibility

42 volunteers have been recruited in this experiment. The sample size is comparable with the reported studies for COVID-19 detection[21–36]. 21 volunteers were positive as the experimental group, 21 healthy volunteers were negative as the control group. This experiment is aimed at the detection of exhaled SARS-CoV-2 by breath and blow. Whether the symptomatic or asymptomatic infected persons can emit SARS-CoV-2 by respiratory activities[5,6]. Therefore, any infected subjects can be the detection objects regardless of the age and gender. The age and gender are random when we recruit the volunteers. For each test subject, we carried out seven tests (0 min, 5 min, 10 min, 15 min, 20 min, 25 min, 30 min) for breath and six tests (0 min, 1 min, 2 min, 3 min, 4 min, 5 min) for blow. 294 tests for breath and 252 tests for blow are carried out for 42 volunteers in total. The results showed good reproducibility for all the tests. The test results are summarized and analyzed in Fig. 4i, j using the scattered plot. To ensure the infection status and eliminate the false health status, each subject was verified using the commercial rapid test kit before the test. At this point, the positive status and negative status were classified using the commercial rapid test kit. Therefore, the data collection for positive volunteers and negative volunteers are not blinded. The collected data were used to redefine the positive status and negative status by the threshold value (−0.2%) of current change ratio of PIDS. According to the threshold value, the diagnosis accuracy can be calculated at different breath time and blow time. After that, the data of 29 volunteers (not blinded) were used for machine learning training, the data of remained 13 volunteers are used for testing the prediction ability of infection status and symptom severity. At this point, the prediction test is a blind test.

## Machine learning

We performed the train-test-splitting and data shuffling on the collected dataset. The total number of volunteers is 42, and the ratio for testing set is more than 25%, which means we have data of 29 subjects for training and 13 subjects for testing. For each volunteer, we collected 28 features including the age, RR, EBT, vaccination, currents at different time by blowing (0 min, 1 min, 2 min, 3 min, 4 min, 5 min), currents at different time by breath (wearing mask; 0 min, 5 min, 10 min, 15 min, 20 min, 25 min, 30 min), current change ratios at different time by blowing (1 min, 2 min, 3 min, 4 min, 5 min), current change ratios at different time by breath (wearing mask; 5 min, 10 min, 15 min, 20 min, 25 min, 30 min). Considering that most of the features showed low correlations to the positive diagnosis and contributed less to the classification, we selectively adopted 7 features as input to the model, including the age, RR, EBT, vaccination, current at 30 min by breath (wear the mask), current change ratios at 5 min and 30 min by breath (wear the mask).

## Reporting summary

Further information on research design is available in the Nature Portfolio Reporting Summary linked to this article.

# Data availability

All data supporting the findings described in this manuscript are available in the article and the Supplementary Information. Source data are provided with this paper.

## Code availability

The machine learning data used in this study are available in the Github at https://github.com/veracoding/Demo.git.

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

## Acknowledgements

This work was supportted by the National Natural Science Foundation of China (Grants No. 62122002), the City University of Hong Kong (Grants No. 9667221, 9680322 and 9678274), in part of the InnoHK Project on Project 2.2—AI-based 3D ultrasound imaging algorithm at Hong Kong Centre for Cerebro-Cardiovascular Health Engineering (COCHE). Prof. L.C. thanks the funding support of National Natural Science Foundation of China (Grants Nos. 32071407 and 62003023), National Key Research and Development Program of China (Grant No. 2022YFB3205600, 2023YFC2415900). Prof. J.H. thanks the funding support of the Government of the Hong Kong Special Administrative Region (COVID1903010, T-11-709/21-N).

## Author contributions

H.L., X.Y. and L.C. conceived the idea. H.L. performed the experiments, processed the data, drew the figures, and prepared the manuscript. H.G. and Y.D. prepared the pseudovirions and live virions. H.L. carried out the test of spike proteins and pseudovirions. Y.D. tested the live virions in P3 lab. H.L. and T.H.W. carried out the test of positive and negative volunteers. J.Z. designed the circuit and developed the apps. H.L., Y.W, L.L. and R.S. conducted the COMSOL simulation. H.L. and H.L.J. carried out the machine learning of the volunteer's data. X.H., Z.G., Y.H., Z.C., W.P., J.Y.L., H.C., S.J., M.W., Y.L., D.L., J.L., G.X., B.Z., Y.G., J.S., C.K.Y. assisted in the large-scale preparation of PIDS when test the volunteers. H.J., H.L., H.C. carried out the machine learning. X.H., B.Z. and J.L. assisted in 3D printing of PIDS. X.H., Z.G., G.X., B.Z., J.Y.L., J.S. and Z.C. assisted in characterization and demonstration of PIDS. All authors discussed the experimental data and results. H.L., X.Y., L.C. and J.H. wrote the paper.

## Competing interests

The authors declare no competing interests.

## Additional information

¹Department of Biomedical Engineering, City University of Hong Kong, Hong Kong 999077, China. ²Beijing Advanced Innovation Center for Biomedical Engineering, School of Biological Science and Medical Engineering, Beihang University, 100083 Beijing, China. ³Key Laboratory of Quantitative Synthetic Biology, Shenzhen Institute of Synthetic Biology, Shenzhen Institutes of Advanced Technology, Chinese Academy of Sciences, Shenzhen 518055, China. ⁴School of Biomedical Sciences, Li Ka Shing Faculty of Medicine, University of Hong Kong, Hong Kong 999077, China. ⁵Hong Kong Centre for Cerebro-Cardiovascular Health Engineering, Hong Kong 999077, China. ⁶College of Engineering, Peking University, 100871 Beijing, China. ⁷Department of Laboratory Medicine, Med+X Center for Manufacturing, West China Precision Medicine Industrial Technology Institute, Department of Liver Surgery, Department of Pathology, West China Hospital, Sichuan University, Chengdu 610041 Sichuan, China. ⁸ Clinical Oncology Center, Shenzhen Key Laboratory for cancer metastasis and personalized therapy, The University of Hong Kong-Shenzhen Hospital, Shenzhen, China. ⁹Guangdong-Hong Kong Joint Laboratory for RNA Medicine, Sun Yat-Sen University, Guangzhou 510120, China. ¹⁰School of Biomedical Engineering, Research and Engineering Center of Biomedical Materials, Anhui Medical University, Hefei 230032, China. ¹¹These authors contributed equally: Hu Li, Huarui Gong, Tsz Hung Wong, Jingkun Zhou, Yuqiong Wang. ✉e-mail: huwenchuang@wchscu.cn; jdhuang@hku.hk; lingqianchang@buaa.edu.cn; xingeyu@cityu.edu.hk

