## [Peer Review File · Nature Communications]

REVIEWER COMMENTS

Reviewer #1 (Remarks to the Author):

Wireless, battery-free, multifunctional integrated bioelectronics for respiratory pathogens monitoring and severity evaluation

This work explains designing and testing a wireless and battery-free systems for respiratory detection of SARS-CoV-2 and its variant, including Omicron and Delta. The detection unit includes an immune biosensor (IBS), respiration biosensor (RBS), and temperature biosensor (TBS) connected to an integrated NFC circuit. The IBS consists of a microchannel for collecting blow and breath samples and an antibody-spiked graphene on gold electrodes. The author suggested that the novelty of their work is rapid sample collection using breath and bow by creating a microchannel with PBS liquid, which makes an air-liquid interface to continuously collect the exhaled viral droplets by dissolving them at the interface.

I think this work can be published in Nature Communication after the following revisions:

1- In the abstract line 39-41, the authors reported on SARS-CoV-2 detection and symptom severity “for diagnosing SARS-CoV-2 infection and symptom severity just by breath and blow within one minute”, this is while for SARS-CoV-2 detection it takes 1 min in blow and 5 mins for the breath test. This should be modified. This time also has a +50s for reading current signals at 50s.

2- When simulating the pressure with 13 inlet numbers which caused excess pressure in the PBS chamber, the PBS should have been blown away from the chamber. Still, it is not clear from the simulation in the Supporting Information Figure. 7. this is also stated in the manuscript line 173-175, but there is no confirmation. In addition, when the outlet length increases (Figure 2.i), the pressure increases, but the airflow decreases at 3mm, and it plateaus after. But in line 178, the author wrongly stated a decrease in the airflow by increasing outlet length.

3- Figure 2.k and Supporting information Fig. 12 show that the airflow decreases by increasing the output width. According to the Supporting information, Fig.11 caption narrow width causes a slower airflow but wider width speeds up the airflow. How this can be explained.

4- To calculate speeds of breath and blow using an atomizer, is there a controlled environment for performing the test (such as a box, ...). In addition, the human breath test was performed using a mask, what was the controlling environment for performing blow tests in human testing. How much this affects the collection and viral capturing speed at the air-liquid interface?

5- What was the temperature for incubation of the Antibody on the graphene surface?

6- It is unclear how bare graphene electrodes can monitor the breath with time (respiratory rate).

7- When all the units are encapsulated with PDMS, it is not clear how PBS liquid was injected in the bionic microchannel, is it before encapsulation or after, and how much PBS is stable during time, is there any leakage or evaporation problems. Does this affect the sensitivity of the biosensor?

8- Is the IBS reusable and stable over time? Is the binding of SARS-COV-2 on the graphene electrode uniform since the mechanism is dissolving in the PBS liquid at the interface, and probably more antigens will be captured on this side of the electrode.

9- In the reported data, why the initial current at the beginning (Time=0) is not similar in the Fig. 4 (e, f, g, h)? This is while a similar Antibody (Ab 1) was used for them. This also occurs for supporting information in Fig. 17 and 18 (a, b, c, d) where initial current is not similar at zero condition.

10- What is the reason for current intensity (amplitude) and rhythm variations in the breath cycles of some cases in breath curves of negative cases in Supporting Information Fig. 23 (such as case 5, 6, 17, 18, this is more for positive cases in Fig. 24 .

11- In Fig 5.e and 5.f, the number of no-fever cases was 13, according to the supporting information Table. 4, only 8 data was shown in Fig. 5e and 9 in Fig. 5f, in the no-fever region, and this caused inaccurate determination of PR and EBT ranges for the no-fever section.

Reviewer #2 (Remarks to the Author):

In this manuscript, Li et al introduce a wireless, battery-free system to monitor respiratory pathogen monitoring. This pathogenic infection diagnosis system can diagnose SARS-CoV-2 infection and symptom severity by breathing and blowing within 5 and 1 minutes with high accuracy among 42 subjects assisted by machine learning. It can be easily integrated with other wearable or portable electronic platforms to realize at-home or public monitoring as a complementary diagnostic tool. The authors should address the following comments before publication.

1. The performance of the IBS is highly dependent on the response current, how about the variations and reproducibility of the baseline current for the batch-to-batch sensors? Do they have similar sensitivity and show a similar current level to the PBS?

2. For positive samples, the concentration of the virus is dependent on the current change ratio, I have concerns about the reliability of such IBS, is there any drift during the working process, especially under the unknown breath/blow condition?

3. For the IBS, the spike antibody is modified on the graphene electrode, how about the sensor performance after long-time storage considering its potential public usage?

4. The paper claimed biomarker identification using machine learning for respiratory detection. When food and drug intake is present, will the ML results be affected? Is ML capable of distinguishing features during events such as food intake? It is important to evaluate the performance of the device in terms of its ability to differentiate biomarkers from normal human activities.
5. The author claimed a varying accuracy for 42 participants. What is the overall accuracy?
6. The naïve Bayes algorithm assumes that the independence of features holds true – it is therefore needed to examine all feature correlations and their independence. However, such an assumption seems flawed in that some features selected shown in Figure 5h has high correlations, which makes applying such an algorithm questionable. Compared with parametric models such as SVM and neural network, does the naïve Bayes algorithm performs better?
7. The details of the ML are not given in the method. Did the author perform train-test-split for ML? Given there are 42 participants, how many data points were collected to train and evaluate the model? In Figure 5g, only 21 cases were shown but 27 variables were used. The mismatch of data size and model complexity will harm the model generalization. In Figure 5i, only 13 cases were evaluated, and this amount is inconsistent with the 21 or 42 participants. It is suggested to clarify these confusions.
8. In Figure 4i, the authors defined the “positive” sample with a current ratio of -0.2% as the threshold value, however, considering the potential motion artifact, signal interference, and sensor drift, how to make sure the accuracy and long-term stability.
9. In Figure 5d, for the temperature recording, there are several regular peaks on ambient temperature waveforms before and after breathing, please comment.
10. Please add scale bars for all the optical images.

Reviewer #3 (Remarks to the Author):

The manuscript titled "Wireless, battery-free, multifunctional integrated bioelectronics for respiratory pathogens monitoring and severity evaluation" by Yu et al presents a system capable of rapidly detecting SARS-CoV-2 infection through breath and blow system. The technology is showcased using various devices such as smartwatches, face masks, wristbands, and mini kits. Note that this technology has not been tested for mutated variants, but rather for the original SARS-CoV-2 strain. I think the utility of this technology is currently limited as there are already numerous tools available for monitoring SARS-CoV-2 infection at home, such as home tests based on lateral flow assays. As a result, this research fails to address the broader problem as it has focused on the wrong disease. This said, I like the extend of the technological development which I think will be of interest to the broader community. My only suggestion would be to explore detection accuracy without breath temperature and compare to the current results. I don't think it is a viable feature for classification.

Responses to comments of Referee

Comments from Referee #1

Summary Comment: Wireless, battery-free, multifunctional integrated bioelectronics for respiratory pathogens monitoring and severity evaluation. This work explains designing and testing a wireless and battery-free systems for respiratory detection of SARS-CoV-2 and its variant, including Omicron and Delta. The detection unit includes an immune biosensor (IBS), respiration biosensor (RBS), and temperature biosensor (TBS) connected to an integrated NFC circuit. The IBS consists of a microchannel for collecting blow and breath samples and an antibody-spiked graphene on gold electrodes. The author suggested that the novelty of their work is rapid sample collection using breath and bow by creating a microchannel with PBS liquid, which makes an air-liquid interface to continuously collect the exhaled viral droplets by dissolving them at the interface. I think this work can be published in Nature Communication after the following revisions.

Our response: We appreciate the reviewer's interest in our system for viral infection diagnosis and thank the reviewer for the positive comments on the novelty of our work. We have carefully addressed the issues raised by the reviewer and revised our manuscript accordingly.

Comment 1: In the abstract line 39-41, the authors reported on SARS-CoV-2 detection and symptom severity “for diagnosing SARS-CoV-2 infection and symptom severity just by breath and blow within one minute”, this is while for SARS-CoV-2 detection it takes 1 min in blow and 5 mins for the breath test. This should be modified. This time also has a +50s for reading current signals at 50s.

Our response: We sincerely thank the reviewer for this comment. In the revised manuscript, we updated the description to make it more clear as “...for diagnosing SARS-CoV-2 infection and symptom severity within 110 s (i.e., 1 min and 50 s) by blowing, and 350 s (i.e., 5 min and 50 s) by breath, respectively. The accuracies reached to 100 % and 92 % for evaluating the infection and severity of 42 participants, respectively.” The comments can be found on Page 2, Lines 41- 43.

Modifications: On Page 2, Lines 41-43, we revised the descriptions of “... for diagnosing SARS-CoV-2 infection and symptom severity within 110 s (i.e., 1 min and 50 s) by blowing, and 350 s (i.e., 5 min and 50 s) by breath, respectively. The accuracies reached to 100 % and 92 % for evaluating the infection and severity of 42 participants, respectively.”

Comment 2: When simulating the pressure with 13 inlet numbers which caused excess pressure in the PBS chamber, the PBS should have been blown away from the chamber. Still, it is not clear from the simulation in the Supporting Information Figure. 7. this is also stated in the manuscript line 173-175, but there is no confirmation. In addition, when the outlet length increases (Figure 2.i), the pressure increases, but the airflow decreases at 3mm, and it plateaus after. But in line 178, the author wrongly stated a

decrease in the airflow by increasing outlet length.

Our response: We thank the reviewer for the insightful and detailed comments. The simulation results in Supporting Information Figure 7 were conducted for calculating the pressure in the symmetrical stable condition. All the airflow velocities in the inlets are 0.01 m/s. In the revised manuscript, we also added the simulation in the asymmetrical transient condition to show the liquid change in the chamber (Figure R1). We simulated the volume fractions of liquid in the chamber with different velocities and compared the air-liquid interface at the same time (0.3 s). A faster velocity tends to cause a higher pressure in the chamber. A higher pressure tends to fluctuate the air-liquid interface. The excessive pressure tends to blow away the PBS from the chamber (Figure R2). In addition, to support the simulation results, we also did the characterizations of the liquid state with the asymmetrical gentle blow, strong blow and excessive blow (Figure R3).

Modifications: On Page 6, Lines 173-174, we modified the text as “However, excessive pressure tends to blow the PBS away from the chamber (Supplementary Figs. 9, 10 and 11).”

On Page 6, Lines 176-178, we added the texts of “The pressure increased with the outlet length. While the airflow velocity decreased first when the outlet length was less than 7 mm, and then it reached the plateaus when the outlet length was longer than 7 mm.”

On pages 24, 25 and 26 in the supporting information, we added supplementary Fig. 9, Fig. 10 and Fig. 11, respectively.

Figure R1. Simulation of volume fraction of PBS with different airflow velocities in an asymmetrical transient condition. With the velocity increased gradually, some of the liquids were beyond the threshold value (0.5), the air-liquid interface lowered gradually. This part of liquid can be regarded as disappeared. **a** Primary state of the liquid without blow. All the airflow velocities are 0 m/s. **b-d** The airflow velocities of the upper left two inlets are 0.01 m/s, 0.1 m/s and 0.5 m/s, respectively. The airflow velocities of other inlets are all 0.01 m/s. The results are captured at 0.3 s.

Figure R2. Simulation of the pressure with different airflow velocities in transient conditions. With the velocity increased gradually, the pressure in the chamber also increased gradually. **a** Primary state of the liquid without blow. All the airflow velocities are 0 m/s. **b-d** The airflow velocities of the upper left two inlets are 0.01 m/s, 0.1 m/s and 0.5 m/s, respectively. The airflow velocities of other inlets are all 0.01 m/s. The results are captured at 0.3 s.

Figure R3. Liquid change in chamber with different strength of blow. The microchannel has 13 inlets and 1 outlet around the chamber. **a** The air-liquid interface kept stable with the gentle blow. **b** The air-liquid interface showed a fluctuated interface with a strong blow. **c** The liquid was blown away with an excessive blow. Scale bar: 1 cm.

Comment 3: Figure 2k and Supporting information Fig. 12 show that the airflow decreases by increasing the output width. According to the Supporting information, Fig.11 caption narrow width causes a slower airflow but wider width speeds up the airflow. How this can be explained.

Our response: We thank the reviewer for the comment. We apologize for the confusion caused from the caption. The original caption of supporting Fig. 11 was that “The pressure decreased with the outlet widths. The narrow width of the outlet means a smaller outlet volume. It induces a slower emission of air from the chamber, which in turn increases the pressure of chamber. The wide width of the outlet means a larger outlet volume. It induces a faster emission of air from the chamber, which in turn decreases the chamber pressure.”.

We would like to clarify that the air emission volume is not equal to the airflow velocity. The slower emission of air from the chamber means a slower air emission volume, not a slower airflow velocity in the outlet. That’s to say, the narrow outlet causes a slower air emission volume, not a slower airflow velocity in the outlet. While the faster emission of air from the chamber means a faster air emission volume, not a faster airflow velocity in the outlet, and therefore the wider outlet causes a faster air

emission volume, not a faster airflow velocity. To make the description more accurate, we made the following modifications on pages 29 and 30 in the revised supporting information.

Modifications: On page 29 in supporting information, the original supplementary Fig. 11 was revised as supplementary Fig. 14. We revised the captions of supplementary Fig. 14 as “Simulation of the air-liquid interface pressure with different outlet widths in the symmetrical stable condition. The airflow velocity of inlets was set as 0.01 m/s. The pressure decreased with the outlet widths. When the outlet height (1mm), outlet number (9) and outlet length (7 mm) were fixed, the narrow and wide widths meant the smaller and larger outlet volumes, respectively. The narrow outlet caused a smaller emission volume at the same airflow velocity, which in turn retained more air in the chamber within the same time and further increased the chamber pressure. The wide outlet induced a larger emission volume at the same airflow velocity, which in turn retained less air in the chamber and further decreased the pressure in chamber.”

On page 30 in supporting information, the original supplementary Fig. 12 was revised as supplementary Fig. 15. We revised the captions of supplementary Fig. 15 as “Simulation of airflow velocity at the outlet end with different outlet widths in the symmetrical stable condition. The airflow velocity of inlets was set as 0.01 m/s. The airflow velocity at the outlet end decreases with the outlet width. When the outlet height (1mm), outlet number (9) and outlet length (7 mm) were fixed, if the same volume of air flowed into the outlet within the same time, it will cause a faster airflow velocity in the narrow outlet and slower airflow velocity in the wide outlet.”

Comment 4: To calculate speeds of breath and blow using an atomizer, is there a controlled environment for performing the test (such as a box, ...). In addition, the human breath test was performed using a mask, what was the controlling environment for performing blow tests in human testing. How much this affects the collection and viral capturing speed at the air-liquid interface?

Our response: We thank the reviewer for this comment, the controlled environmental conditions were set for both experiments and simulation, which include: (1) The simulation test, blow test and breath test were carried out at 25 °C; (2) The temperature of PBS was 25 °C. (3) The distance between the human mouth and microchannel inlets was 1 cm. The distance between the atomizer and microchannel inlets was 0 cm.

In revision, we added more comprehensive experiments for this purpose, where a tube (which provides an enclosed space like the mask or box) as the controlling environment is introduced to study its influence on the collection and viral capturing speed at the air-liquid interface. The height, diameter and thickness of the tube were 6 cm, 2.2 cm, and 1 mm, respectively. As shown in Figure R4 (Figure R4 was added as supplementary Fig. 17 in the revised supporting information), the collection speed decreased with the distance from 0 cm to 2 cm (Figure R4a to R4c), which can be attributed to the decreased airflow velocity with the distance. When a tube was added as the controlling condition (Figure R4d), the collection speed was slower than that without the tube at 0 cm distance in Figure R4a, but faster than that without tube at 2

cm distance in Figure 4c. The results indicate that the enclosed space provided by the spray nozzle of the atomizer (Figure R4a) and the tube (Figure R4d) help to collect the aerosols, which could be attributed to the higher pressure in the enclosed space. Meanwhile, the longer the distance, the slower the collection speed.

Figure R4. Collection simulation of aerosols using an atomizer. **a-c** The microchannel was placed at different positions. The distances between the atomizer and microchannel were (a) 0 cm, (b) 1 cm, and (c) 2 cm. The collection speed decreased with the increased distance. **d** A tube was added as a controlling condition to test the collection speed of aerosols. The distance between the tube and microchannel was 0 cm.

As shown in Figure R5 (Figure R5 was added as supplementary Fig. 18 in the revised supporting information), compared with the human blow data in original manuscript, the collection speed with the tube by mouth blow also became slower than that without the tube. The results can be attributed to the temperature difference between human blow and the environment. Considering that the airflow velocity by mouth blow is much faster than that of atomizer, the velocity attenuation in the tube (6 cm in length) can be neglected. However, the temperature of mouth blow was higher (about 32.5 °C) than the environment temperature (25 °C). The temperature difference in the open environment without tube help to condense the exhaled aerosols and accelerate its collection. On the contrary, the tube confined the airflow in the enclosed space and reduced the temperature difference. It is unfavorable for condensing and collecting the aerosols. This could be the main reason for the slower collection speed in an enclosed space.

The results showed that the direct blow by mouth in an open environment in our experiment is the best method to collect the aerosol samples. The distance between mouth and inlets was 1 cm.

Figure R5. Collection speed of the aerosols with mouth blow. The tube was added on the microchannel and covered the inlets. The mouth directly blew the upper end of the tube. The distance between mouth and the upper end of the tube was 1 cm. The blow frequency was 20 times per minute. The time was about 3 seconds for each blow cycle.

Modifications: On page 32 in supporting information, we added the supplementary Fig. 17. We added the related texts of “We added a tube as the controlling environment to explore its influence on the collection and viral capturing speed at the air-liquid interface. The height, diameter and thickness of the tube were 6 cm, 2.2 cm, and 1 mm, respectively. The environment temperature and ink temperature are all 25 °C. As shown in supplementary Fig. 17, the collection speed decreased with the distance from 0 cm to 2 cm (Fig. 17a to 17c), which can be attributed to the decreased airflow velocity with the distance. When a tube was added as the controlling condition (supplementary Fig. 17d), the collection speed was slower than that without the tube at 0 cm distance in supplementary Fig. 17a, but faster than that without tube at 2 cm distance in supplementary Fig. 17c. The results indicated that the enclosed space provided by the spray nozzle of the atomizer (supplementary Fig. 1a) and the tube (supplementary Fig. 17d) help to collect the aerosols, which could be attributed to the higher pressure in the enclosed space. Meanwhile, the longer the distance, the slower the collection speed.”.

On page 33 in supporting information, we added supplementary Fig. 18. We added related texts of “Compared with the mouth blow without the tube in the manuscript (human blow, Fig. 2o), the collection speed with the tube by mouth blow became slower. The results can be attributed to the temperature difference between human blow and the environment. Considering that the airflow velocity by mouth blow is much faster than that of atomizer, the velocity attenuation in the tube (6 cm in length) can be neglected. However, the temperature of mouth blow is higher (about 32.5 °C) than the environment temperature (25 °C), the temperature difference in the open environment without tube help to condense the exhaled aerosols and accelerate its collection. On the contrary, the tube provided an enclosed space that confined the airflow in the tube and reduced the temperature difference. It is unfavorable for condensing and collecting the aerosols. This could be the main reason for the slower collection speed in an enclosed space.”.

On page 7, Lines 199-204 in the manuscript, we added the texts of “The collection speed decreased with the distance between the atomizer and microchannel both in an enclosed space and an open space, which could be attributed to the reduced airflow velocity in a longer distance. For the blow test, the temperature difference between the

exhaled aerosols and environment in the open space was more beneficial to collect the aerosols than the enclosed space (Supplementary Fig. 17 and Fig. 18).”

On page 20, Lines 581-583 in the manuscript, we added the texts of “... at 25 °C controlled by the household air conditioner.”

On page 19, Lines 588-589 in the manuscript, we added the texts of “The distance between mouth and the microchannel was 1 cm.” “The time was 3 s for each blow cycle.”.

Comment 5: What was the temperature for incubation of the Antibody on the graphene surface?

Our response: We thank the reviewer for this comment. The incubation of the antibody on graphene surface was carried out at room temperature (25 °C), which has been added in experimental section in the revised manuscript.

Modifications: On page 16, Lines 480-481 in the manuscript, we added the text of “The spike antibody (10 µl, 250 µg/ml) of SARS-CoV-2 was drop-cast onto the graphene and incubated for 3 h at room temperature (25 °C).”

Comment 6: It is unclear how bare graphene electrodes can monitor the breath with time (respiratory rate).

Our response: We appreciate the reviewer for this comment and apologize for any confusion caused by our unclear description. The graphene resistance increases or decreases upon the adsorption or desorption of aerosols on its surface, respectively. Validation and qualitative studies can be seen in recent works ^{1,2}. According to the working mechanism, when a constant voltage was applied on the RBS module, the output current decreased and increased periodically with the breath. The collected signals were wirelessly transmitted to the mobile phone by NFC technique. The peak detection algorithm was applied to identify the crest locations of each respiratory wave. By counting the number of peaks within a given time window, the respiratory rate can be derived and defined as the number of breaths taken per minute (Figure R6 and Figure R7).

Figure R6. Respiratory rate (RR) calculation of a regular breath curve. There are 14 peaks from 34 s to 86.2 s. The RR can be calculated as $RR = \frac{60\text{ s}}{(86.2-34)\text{ s}} \times 14 \approx$

16 min^{-1} .

Figure R7. Respiratory rate (RR) calculation of an irregular breath curve. There are 12 peaks from 31.6 s to 79.78 s. The RR can be calculated as $RR = \frac{60}{(73.6-31.6)s} \times 12 \approx 17 \text{ min}^{-1}$.

Modifications: On pages 11-12, Lines 340-343 in the manuscript, we added the texts of “Fig. 5b showed the working mechanism of RBS by adsorption or desorption of aerosols on graphene that increased or decreased the surface resistance, respectively. When a constant voltage was applied on the RBS module, the output current decreased and increased periodically with the breath.”.

On page 12, Lines 345-349 in the manuscript, we added the texts of “The collected signals were wirelessly transmitted to the mobile phone by NFC technique. The peak detection algorithm can be applied to identify the crest locations of each respiratory wave. By counting the number of peaks within a given time window, the respiratory rate can be derived and defined as the number of breaths taken per minute (Supplementary Figs. 28 and 29).”.

On pages 45 and 46 in supporting information, we added supplementary Fig. 28 and supplementary Fig. 29, respectively.

Reference:

1. Li, Y. et al. Graphdiyne-based flexible respiration sensors for monitoring human health. *Nano Today* 39, 101214 (2021).
2. Zhen, Z. et al. Formation of uniform water microdroplets on wrinkled graphene for ultrafast humidity sensing. *Small* 14, 1703848 (2018).

Comment 7: When all the units are encapsulated with PDMS, it is not clear how PBS liquid was injected in the bionic microchannel, is it before encapsulation or after, and how much PBS is stable during time, is there any leakage or evaporation problems. Does this affect the sensitivity of the biosensor?

Our response: We thank the reviewer for this comment. The PBS liquid was injected in the bionic microchannel by an injector through the outlet after the encapsulation (see Figure R8 below). The volume of 40 μL PBS was stable during time. The microchannel

was solidly stuck on the substrate with A+B glue (epoxy resin). There is no liquid leakage occurred during the experiment, which can be attributed to the reliable encapsulation and hydrophobic property of PDMS (see Figure R9 below). The hydrophobic property means a low surface energy and makes the PDMS surface difficult to be wetted. The PBS is confined in the narrow space of chamber and difficult to flow around even if it faces to different directions (see Figure R10 below).

For the evaporation concern, the device after PBS injection was stored at $-20\text{ }^{\circ}\text{C}$ to protect the antibodies' activities for further usage (can be stored 12 months, information provided by the antibody supplier). There is no observable evaporation and leakage occurred during the storage condition (see Figure R11 below). When we tested the device for breath in mask or blow with mouth, we found no evaporation that could be attributed to the saturated exhaled aerosols by breath and blow. The storage at $-20\text{ }^{\circ}\text{C}$ showed stable performance and has not shown influence on the sensitivity in our test for the negative and positive participants (see supplementary information Fig. 25 and Fig. 26). Additionally, we also tested the performance of the biosensor with different volumes of PBS from $10\text{ }\mu\text{L}$ to $40\text{ }\mu\text{L}$, the biosensor showed stable current without obvious variation (see Figure R12 below), which proved that the liquid volume would not affect the sensitivity of the biosensor.

Figure R8. The red ink (or PBS solution) was injected into the chamber from the outlet using an injector. a Picture of the microchannel before injection. b Picture of the microchannel with a little red ink in the chamber injected by the injector. c Picture of the microchannel with proper amounts of red ink. Scale bar: 1 cm.

Figure R9. **d** Surface morphologies and **e** contact angle of PDMS. The contact angle is about 100° , proving the hydrophobic property of PDMS. Figure R9 d and e come from Figure 2 d and e in the original manuscript.

Figure R10. The hydrophobicity of PDMS contributed to keeping the air-liquid interface stable in different directions. **a** Up. **b** Left. **c** Down. The PBS solution in the chamber keeps stable without flowing out at any directions. Scale bar: 1 cm.

Figure R11. The sample was stored at -20°C in the refrigerator. There is no observable evaporation and leakage occurred for 2 months.

Figure R12. Real-time current curves recorded by dropwise adding 10 μL of PBS on the biosensor. The total volume of 40 μL was added on the biosensor. The current kept stable in different volumes of PBS proving that the liquid volume would not affect the sensitivity of the biosensor.

Modifications: On pages 56 and 57 in the supporting information, we added the supplementary Fig. 33 and Fig. 34, respectively.

On page 16, Lines 494-499 in the manuscript, we added the descriptions of “The IBS module was covered with a bionic PDMS microchannel using A+B glue (epoxy resin). Then the PBS (30 μL -40 μL) was injected into the chamber (Supplementary Fig. 37) to immerse the graphene and protect the antibody activity. The samples were stored in the refrigerator at $-20\text{ }^{\circ}\text{C}$ for further usage. There is no observable evaporation and leakage occurred during the storage (Supplementary Fig. 38).”

Comment 8: Is the IBS reusable and stable over time? Is the binding of SARS-COV-2 on the graphene electrode uniform since the mechanism is dissolving in the PBS liquid at the interface, and probably more antigens will be captured on this side of the electrode.

Our response: Thanks to the reviewer for the insightful questions. For the positive individuals, it is not recommended for reusable applications because that the IBS will be contaminated by the exhaled virus including the surface and inner space. The reuse has the risk of reinfection and spreading virus around. For the negative individuals, it can be reasonable to think that the IBS is reusable under suitable conditions, for example, storing the device in refrigerator at $-20\text{ }^{\circ}\text{C}$. In our practical tests for the negative and positive individuals, we first prepared the integrated device in large scale and then stored them in the refrigerator at $-20\text{ }^{\circ}\text{C}$ in our lab. Appropriate amount of PBS was injected into the chamber through the outlet to cover the graphene and keep it in the liquid condition. The tests of negative and positive participants are completed in 4 months, the IBS showed stable and sensitive performance to the users (see Supplementary Fig. 25 and Fig. 26 in the supporting information). According to the antibody supplier, the spike antibody for SARS-CoV-2 is stable without detectable loss of activity over 1 month at $2\text{ }^{\circ}\text{C}$ - $8\text{ }^{\circ}\text{C}$ and 12 months at $-20\text{ }^{\circ}\text{C}$ to $-80\text{ }^{\circ}\text{C}$.¹ Except the low temperature ($-20\text{ }^{\circ}\text{C}$), we also tested the performance of IBS at room temperature

(25 °C) for three days, the results showed that IBS was reusable for identifying biomarkers under room temperature (see Figure R13 below).

Besides, we agree with that the binding of SARS-CoV-2 virus will be not so uniform since the mechanism is dissolving the virus droplets in the PBS liquid at the interface, and more antigens may be captured on this side of the electrodes. Nevertheless, this will not affect the detection of virus because of the parallel structure of interdigital electrodes. We assume that the interdigital electrode consists of n pairs of parallel electrodes. The graphene was uniformly distributed on the electrodes. Each pair of the electrodes has an equal resistance. That is to say, $R_1 = R_2 = \dots = R_n = R$. The resistance of the whole interdigital electrodes can be calculated by the formula:

$$\frac{1}{R_{whole}} = \frac{1}{R_1} + \frac{1}{R_2} + \dots + \frac{1}{R_n}. \text{ Based on this formula, we can obtain that } R_{whole} = \frac{R}{n}.$$

The changed resistance can be calculated as $\Delta R_{whole} = R_{after} - R_{before} = \left(\frac{R}{n} + \frac{\Delta R}{n_1}\right) -$

$$\frac{R}{n} = \frac{\Delta R}{n_1}. R_{after} \text{ and } R_{before} \text{ are the resistances of graphene after and before it bound with}$$

viruses. That is to say, $\Delta R_{whole} = \frac{\Delta R}{n_1}$. Here, ΔR is the changed resistance caused by

the virus binding on the n_1 pairs of electrodes. n_1 is the number of electrodes that bond with virus whether on the upper side or the lower side. From this formula, $\Delta R_{whole} =$

$\frac{\Delta R}{n_1}$, the changed resistance of IBS is related to the pairs of electrodes that bound with

viruses, but not the positions of the electrodes.

Figure R13. Identification ability of IBS to SARS-CoV-2 spike protein under room temperature. The concentrations of BSA and spike protein were 500 fg ml^{-1} . BSA protein was used for verifying the specificity and the original current value before every test. IBS can still identify the SARS-CoV-2 spike proteins proving the reusable ability during the practical applications. The samples were stored in PBS solution at room temperature for 3 days.

Modifications: On page 15, Lines 433-438 in the manuscript, we added the texts of “The last but not the least, the IBS won’t be reusable after testing the positive cases to

avoid virus transmission. While the IBS is still reusable if the tests showed negative results. Furthermore, the IBS can be still active by storing it at room temperature for three days (supplementary Fig. 35) or in the refrigerator at -20 °C for several months.”.

On page 15, Lines 457-460 in the manuscript, we added the texts of “Considering that the dissolution of viral aerosols was initiated from the air-liquid interface, the parallel design of interdigital microelectrodes helps to capture viruses on multiple pairs of microelectrodes and weaken the influence of the uneven diffusion and binding of virus on graphene electrodes.”.

On page 19, Lines 578-579 in the manuscript, we added the texts of “We did the human tests for 4 months. All the PIDS stored in the refrigerator at -20 °C can work normally during the period of human test.”.

Reference:

1. <https://cn.sinobiological.com/antibodies/cov-spike-40150-r007>

Comment 9: In the reported data, why the initial current at the beginning (Time=0) is not similar in the Fig. 4 (e, f, g, h)? This is while a similar Antibody (Ab 1) was used for them. This also occurs for supporting information in Fig. 17 and 18 (a, b, c, d) where initial current is not similar at zero condition.

Our response: We thank the reviewer for the insightful comments. In Fig. 4 (e, f, g, h) in the manuscript and supplementary Figs. 17 and 18 (a, b, c, d) (which are Fig. 22 and Fig. 23 in the revised supporting information), these tests were all carried out using the antibody 1 with different devices. These phenomena also happened in supplementary Fig. 25 and Fig. 26 in the revised supporting information for positive and negative human tests. The initial current at the beginning (Time = 0) is determined by the resistance of graphene on the interdigital electrodes. During the preparation of IBS module, the graphene dispersion in water (500 μ L) was sprayed on the interdigital electrodes at 140 °C. The graphene dispersion was dispersed in water using the surfactant (SDBS, sodium dodecyl benzene sulfonate) by the manufacturer, which is insulated and non-conductive. The surfactant was mixed with graphene on the interdigital electrodes after the spraying. It was removed by immersing the samples in deionized water for 12 hours and dried on a hotplate at 80 °C. The immersion step can cause the desorption of some graphene and fall off from the electrodes, which in turn resulted in the resistance difference of samples. In addition, the subsequent modification of PBASE, antibody and BSA on graphene will also cause some resistance difference. These combined actions finally produced different original sample resistances and caused the current difference at the beginning (Time = 0). In our experiment, we adopted the current change ratio (not the absolute current value) to monitor the capture of virus, which eliminated the effect of different baseline current values. The current change ratio was calculated by the formula: $(I-I_0)/I_0 \times 100\%$, where I and I_0 were the measured average stable current and original stable current, respectively.

Modifications: On page 16, Lines 484-492 in the manuscript, we added the texts of

“The cleaning of SDBS together with the modification of PBASE, antibodies and BSA will lead to some difference of the baseline currents between different IBS modules. The baseline currents of IBS module in PBS varied between 160 μ A and 350 μ A. In our experiment, we adopted the current change ratio (not the absolute current value) as sensing marker to monitor the captured virus, which can eliminate the effect of difference in sensing data baselines. The results in the following tests also proved this point that the difference in baseline currents showed no influence on the detection of viral infection (Fig 3e-p; Supplementary Fig. 21, Fig. 22, Fig. 23, Fig. 25 and Fig. 26).”.

Comment 10: What is the reason for current intensity (amplitude) and rhythm variations in the breath cycles of some cases in breath curves of negative cases in Supporting Information Fig. 23 (such as case 5, 6, 17, 18, this is more for positive cases in Fig. 24).

Our response: Many thanks for your insightful questions. The graphene resistance increases and decreases with the adsorption and desorption of aerosols on its surface, respectively. Validation and qualitative studies can be seen in recent works^{1,2}. According to the working mechanism, when a constant voltage was applied on the RBS module, the output current decreased and increased periodically with the breath. The current intensity (amplitude) could represent/reflect the breath depth³⁻⁵, which was decided by the amounts of exhaled aerosols. When the users breathe deeply, more aerosols were exhaled and absorbed on graphene, which caused a large current decrease and increase (large amplitude variation) during the exhalation and inhalation. When the users breathe gently, less aerosols were exhaled and absorbed on graphene, which in turn cause a low current decrease and increase (small amplitude variation) during the exhalation and inhalation. The amplitude variations indicated that the cases 5, 6, 17, 18 have taken the deep breaths and shallow breaths during the breath tests with mask.

The rhythm could represent/reflect the respiratory health⁶⁻⁹. Normally, for the healthy subjects without nasal disease, the amplitude and breath rhythm are similar and regular. However, some of the healthy subjects (negative cases) with rhinitis (nasal obstruction) will cause abnormal breath rhythms within limits. Sometimes the participants showed assisted breath by mouth due to the lack of oxygen. These phenomena occurred more frequently for the infected subjects with nasal obstructions (or runny nose and cough)¹⁰. All these breath differences were sensitively recorded by the respiration biosensor (RBS). In our work, we adopted the respiration rate (RR) and exhaled breath temperature (EBT) as the indicators for diagnosing virus infection. On the other hand, we believe that the difference of current amplitude and breath rhythm can be also potentially adopted as the indirect indicators for comprehensively diagnosing virus infection and symptom severity in future works.

Modifications: On page 12, Lines 355-360 in the manuscript, we added the texts of “For the negative cases, most of the current intensities (amplitude) and rhythms are similar and regular due to the unobstructed breath. Some irregular current intensities and rhythms occurred in the curves for the cases 5, 6, 17, 18 because of the rhinitis (nasal obstruction). This phenomenon is witnessed more frequently among positive

cases with nasal obstructions (runny nose, cough).”.

References:

1. Li, Y. et al. Graphdiyne-based flexible respiration sensors for monitoring human health. *Nano Today* **39**, 101214 (2021).
2. Zhen, Z. et al. Formation of uniform water microdroplets on wrinkled graphene for ultrafast humidity sensing. *Small* **14**, 1703848 (2018).
3. Zhu, P. et al. Cellulose nanofiber/carbon nanotube dual network-enabled humidity sensor with high sensitivity and durability. *ACS Appl. Mater. Interfaces* **12**, 33229-33238 (2020).
4. Pang, Y. et al. Wearable humidity sensor based on porous graphene network for respiration monitoring. *Biosens. Bioelectron.* **116**, 123-129 (2018).
5. Dinh, T. et al. Stretchable respiration sensors: Advanced designs and multifunctional platforms for wearable physiological monitoring. *Biosens. Bioelectron.* **166**, 112460 (2020).
6. Sozansky, J., Houser, S. M.. The physiological mechanism for sensing nasal airflow: A literature review. *International forum of allergy & rhinology.* **4**, 834-838 (2014).
7. Rappai, M. et al. The nose and sleep-disordered breathing: what we know and what we do not know. *Chest* **124**, 2309-2323 (2003).
8. Bartley, J. Nasal influences on breathing. *Recognizing and Treating Breathing Disorders*, 45-50 (2014).
9. Pevernagie, D. A., Meyer M. M. D. & Claeys S. Sleep, breathing and the nose. *Sleep Med. Rev.* **9**, 437-451 (2005).
10. Lovato, A. & De Filippis, C. Clinical presentation of COVID-19: a systematic review focusing on upper airway symptoms. *Ear Nose & Throat Journal* **99**, 569-576 (2020).

Comment 11: In Fig 5.e and 5.f, the number of no-fever cases was 13, according to the supporting information Table. 4, only 8 data was shown in Fig. 5e and 9 in Fig. 5f, in the no-fever region, and this caused inaccurate determination of PR and EBT ranges for the no-fever section.

Our response: Thank you for the insightful comments. We apologize for the confusion caused by the unclear annotations. Fig 5e and 5f in the original manuscript have shown 13 no-fever cases and data points (see the red box blow). There are 7 mild positive cases showed no fever. Their respiration rate (RR) and exhaled breath temperature (EBT) were $20 < RR < 24$ (Fig. 5e) and $33\text{ }^{\circ}\text{C} < \text{EBT} < 34\text{ }^{\circ}\text{C}$ (Fig. 5f), respectively. There are 6 near asymptomatic positive cases showed no fever. Their RR and EBT were $RR \leq 20$ (Fig. 5e) and $\text{EBT} \leq 33\text{ }^{\circ}\text{C}$ (Fig. 5f), respectively.

Figure 5. e, f Statistical analysis of RR and EBT for 42 cases, respectively. **e** Eight positive cases with high fever showed severe symptoms, their respiration rates were $RR \geq 24$. Seven positive cases with no fever showed mild symptoms, their respiration rates were $20 < RR < 24$. Six positive cases with no fever showed near asymptomatic behaviors, their $RR \leq 20$. The twenty-one negative cases showed normal health status, their respiration rates were $RR \leq 20$. **f** Eight positive cases with high fever showed severe symptoms, their exhaled breath temperatures were $EBT \geq 34^\circ\text{C}$. Seven positive cases with no fever showed mild symptoms, their exhaled breath temperatures were $33^\circ\text{C} < EBT < 34^\circ\text{C}$. Six positive cases with no fever showed near asymptomatic behaviors, their exhaled breath temperatures were $EBT \leq 33^\circ\text{C}$. The twenty-one negative cases showed normal health status, their exhaled breath temperatures were $EBT \leq 33^\circ\text{C}$.

Modifications: On page 31 in the manuscript, we added new annotations in Fig. 5e and Fig. 5f. For Fig. 5e, the annotations are “ $RR \geq 24$ severe/high fever, (8 positive cases); $20 < RR < 24$ mild/no fever, (7 positive cases); $RR \leq 20$ near asymptomatic/no fever, (6 positive cases); $RR \leq 20$ (21 negative cases)”. For Fig. 5f, the annotations are “ $EBT \geq 34^\circ\text{C}$ severe/high fever, (8 positive cases); $33^\circ\text{C} < EBT < 34^\circ\text{C}$ mild/no fever (7 positive cases); $EBT \leq 33^\circ\text{C}$ near asymptomatic/no fever (6 positive cases); $EBT \leq 33^\circ\text{C}$ (21 negative cases)”.

Comments from Referee #2

Summary Comment: In this manuscript, Li et al introduce a wireless, battery-free system to monitor respiratory pathogen monitoring. This pathogenic infection diagnosis system can diagnose SARS-CoV-2 infection and symptom severity by breathing and blowing within 5 and 1 minutes with high accuracy among 42 subjects assisted by machine learning. It can be easily integrated with other wearable or portable electronic platforms to realize at-home or public monitoring as a complementary diagnostic tool. The authors should address the following comments before publication.

Our response: We thank the referee for these positive comments and insightful suggestions for improvement. We carefully addressed the issues, as listed below, and we revised our manuscript accordingly.

Comment 1: The performance of the IBS is highly dependent on the response current, how about the variations and reproducibility of the baseline current for the batch-to-batch sensors? Do they have similar sensitivity and show a similar current level to the

PBS?

Our response: Sincerely thanks for your insightful questions. There are two steps that may affect the baseline current during the preparation of IBS. One step is rinsing the IBS samples. The graphene dispersion solution contains the non-conducting surfactant, sodium dodecyl benzene sulphonate (SDBS).¹ After spraying the dispersion solution on the interdigital electrodes, the IBS samples were immersed into deionized water for 12 hours to dissolve and remove the SDBS, which might bring some graphene into water again and influenced the original resistance of IBS^{2,3}. Another step was the modification of non-conducting molecular linker (PBASE), antibodies and BSA on graphene, which might also affect the original resistance of IBS. These two steps caused the difference in baseline current under a constant voltage during the batch-to-batch production. The baseline currents in PBS varied between 160 μ A and 350 μ A due to their different original resistance (Supplementary Figure 25 and 26 in the revised Supporting Information). We have prepared hundreds of IBS samples that showed good reproductive properties in this current range. In our experiment, we adopted the current change ratio (not the absolute current value) to monitor the capture of virus, which eliminated the effect of different baseline current values. The current change ratio was calculated by the formula: $(I-I_0)/I_0 \times 100\%$, where I and I_0 were the measured average stable current and original stable current, respectively.

We statistically calculated the data fluctuation range (i.e., -0.2%) of negative cases to eliminate this influence of baseline current (Figure 4i and j in the manuscript). Based on the threshold value (-0.2%), we redefined the positive and negative cases. When the current change ratio was larger than the threshold value, it was defined as “positive”. When the current change ratio was smaller than the threshold value, it was defined as “negative”. Based on this definition, the batch-to-batch IBS sensors showed similar sensitivity during the tests of human participants in PBS solution (Supplementary Fig. 25 and Fig. 26 in the revised Supporting Information).

Modifications: On page 16, Lines 484-492 in the manuscript, we added the text of “The cleaning of SDBS together with the modification of PBASE, antibodies and BSA will lead to some difference of the baseline currents between different IBS modules. The baseline currents of IBS module in PBS varied between 160 μ A and 350 μ A. In our experiment, we adopted the current change ratio (not the absolute current value) to monitor the capture of virus, which can eliminate the effect of difference in sensing data baselines. The results in the following tests also proved this point that the difference in baseline currents showed no influence on the detection of viral infection (Fig 3e-p; Supplementary Fig. 21, Fig. 22, Fig. 23, Fig. 25 and Fig. 26).”.

References:

1. Nazari, B. et al. Dispersing graphene in aqueous media: Investigating the effect of different surfactants. *Colloid Surface A*. **582**, 123870 (2019).
2. Tran, T. S. et al. Graphene inks for printed flexible electronics: Graphene dispersions, ink formulations, printing techniques and applications. *Adv. Colloid Interfac.* **261**, 41-61 (2018).

3. Li, H. et al. Fabrication and test of adhesion enhanced flexible carbon nanotube transparent conducting films. *Appl. Surf. Sci.* **313**, 220-226 (2014).

Comment 2: For positive samples, the concentration of the virus is dependent on the current change ratio, I have concerns about the reliability of such IBS, is there any drift during the working process, especially under the unknown breath/blow condition?

Our response: Thank the reviewer for the insightful questions. The reactive area of IBS was covered by the PBS solution in the chamber, which avoid the influence of moisture from unknow breath/blow. This point was verified by Figure R14 below. When a sudden breath or blow happened, the current curve of IBS kept stable without drift. Additionally, to eliminate the concerns in practical tests, we firstly tested the response currents of the IBS for 21 negative cases as the control group. The current values showed no drift but tiny fluctuation within limits in repeated tests (see Supplementary Fig. 26 in supporting information), which were statistically calculated and plotted as the scatter chart in Figure 4 i and j in the original manuscript. The fluctuation range formed the threshold value (-0.2%) that was used to differentiate the positive cases from negative cases. Based on this method, we achieved the 100 % accuracy when diagnosing the positive and negative cases by breath and blow for human tests, which proved the stable and reliable performance of IBS under the unknown breath/blow conditions.

For the protein test and virus test, the baseline current and response current curves at low concentrations kept stable and showed no drift (Supplementary Fig. 21 and Fig. 22 in supporting information), which further proved the stable and reliable performance of IBS. However, the current curves (response current, not baseline current) at high concentrations showed drift phenomenon, which was because that the efficient dynamic binding between antibody and antigen enhanced gradually with incubation time.¹ The resistance gradually increased when more antigens were captured and bound with antibodies on graphene, which in turn decreased the response current with time at a constant voltage of 0.2 V. This phenomenon is consistent with the detection mechanism of our experiment, which means that the binding of virus leads to increase of the resistance of graphene and decrease of response current. These results indicate that IBS is reliable for diagnosing the virus infection by breath test and blow test.

Figure R14. Current curve of IBS kept stable when suffered from a sudden breath and

blow, proving the reliability of IBS during the working process, especially under the unknow breath/blow condition.

Figure 4 i and j in original manuscript. Statistic analysis of the current values of positive cases and negative cases by blow test and breath test, respectively. The current change ratio of -0.2 % was adopted as the threshold value to redefine the “positive” and “negative”.

Supplementary Fig. 21 in supporting information. a. Current curves tested for different concentrations (0 fg ml⁻¹ to 1 pg ml⁻¹) of spike proteins. The baseline current (0 fg ml⁻¹) and response current curve at low concentration (0.5 fg ml⁻¹) kept stable and showed no drift (decrease). These two curves overlapped one another. The response current at 1 fg ml⁻¹ showed a slight drift (decrease) due to the binding of virus, which is the limit of detection (LOD) of IBS to spike proteins. When further increase the concentrations, the response current gradually decreased with more specific binding reactions of nonconducting spike proteins. b. Similarly, this phenomenon occurred in the selectivity test of BSA, MERS, SARS-CoV and SARS-CoV-2. The baseline current

curve in PBS overlapped the response current curve of nonreactive BSA and MERS. **c.** The selectivity test of Delta also proved this phenomenon that the baseline current curves in PBS kept stable and overlapped the response current curve of nonreactive Omicron. The drift (decrease) occurred due to the specific binding of reactive Delta. **d.** The selectivity test of Omicron also proved this phenomenon that the baseline current curves in PBS kept stable and overlapped the response current curve of nonreactive Delta. The drift (decrease) occurred due to the specific binding of reactive Omicron. These drifts are not from the sensor itself but from the specific binding of corresponding spike proteins.

Supplementary Fig. 22 in supporting information. Current curves tested for different concentrations (0 fg ml⁻¹ to 1 pg ml⁻¹) of pseudo virus including **a,** WT; **b,** Alpha; **c,** Delta; **d,** Omicron. The baseline current (0 fg ml⁻¹) and response current curves at low concentrations kept stable and showed no drift (no decrease). The curves lower than 0.5 pg ml⁻¹ overlapped with each other. When the pseudovirus concentration was higher than 20 pg ml⁻¹, the response current curves showed drift (see the red arrows) and gradually decreased with more specific binding reactions of pseudovirus. Similarly, this phenomenon also occurred when tested the SARS-CoV-2 variants, for example the Alpha, Delta and Omicron. Compared with the baseline and other current curves at low concentrations, these drifts are not from the sensor itself but from the specific binding of corresponding spike proteins.

Modifications: On page 9, Lines 258-267 in the manuscript, we added the texts of “Additionally, when the concentration of spike protein was lower than the LOD, the response current curves overlapped with the baseline current. When the spike protein was not reactive with the antibody, the response current curves also overlapped with the baseline current (Supplementary Fig. 21a-d). These results indicated that the IBS module was stable without drift during the test. When the protein was reactive with the antibody, the current decreased with time at the high concentration (500 fg/ml in

Supplementary Fig. 21c and d), which was because that the efficient dynamic binding between the antibody and protein was enhanced gradually with incubation time⁵⁰. More spike proteins can bind with the antibodies over time.”

On page 10, Lines 300-301 in the manuscript, we added the texts of “We first tested the negative participants and then statistically analyzed the current change ratios using the boxplot (Fig. 4i, j).”.

On page 10, Lines 303-307 in the manuscript, we added the texts of “The current change ratios of negative cases just fluctuated within a narrow range and no drift occurred during the repeated tests, proving the stable performance of IBS under the breath and blow conditions. Then we tested the positive cases and compared them with the negative cases (Fig. 4i, j).”.

On page 37, In supplementary Fig. 22, we added the captions of “Additionally, when the concentration of pseudo viruses was lower than the LOD (0.5 pg ml⁻¹), the response current curves overlapped with the baseline current, indicating that IBS was stable without drift during the test. When the concentration was very high (e.g., 500 pg ml⁻¹), the response current decrease gradually with time, which was because that the efficient dynamic binding between the antibody and spike protein of pseudo viruses was enhanced gradually with incubation time. More pseudo viruses can bind with the antibody over time.”.

On page 63 in supporting information, we added supplementary Fig. 40. On page 17, lines 499-502, we added texts of “Additionally, the IBS module could keep stable baseline current and showed no current drift under the unknow breath and blow conditions (e.g., sudden breath and blow) (Supplementary Fig. 40).”.

Reference:

1 Reverberi, R. & Reverberi, L. Factors affecting the antigen-antibody reaction. *Blood Transfus* **5**, 227-240 (2007).

Comment 3: For the IBS, the spike antibody is modified on the graphene electrode, how about the sensor performance after long-time storage considering its potential public usage?

Our response: Sincerely thanks for your insightful questions. After we prepared the IBS and injected PBS solution into the chamber, the samples were stored in the refrigerator at -20 °C. Then we recruited and tested the positive and negative participants in the following 4 months. The results of human tests in Supplementary Fig. 25 and Fig. 26 showed that the IBS kept stable and was qualified for identifying the positive cases and negative cases within 4 months. Additionally, according to the antibody supplier, the spike antibody could keep stable without detectable loss of activity over 1 month at 2 °C-8 °C and 12 months at -20 °C to -80 °C.¹ Except the low temperature (-20 °C), we also tested the performance of IBS at room temperature (25 °C) for three days, the results showed that IBS was competent and reusable for identifying biomarkers after 3 days of storage under room temperature (see Figure R14 below).

Figure R14. Identification ability of IBS to SARS-CoV-2 spike protein under room temperature. The concentrations of BSA and spike protein were 500 fg ml⁻¹. BSA protein was used for verifying the specificity and the original current value before every test. IBS can still identify the SARS-CoV-2 spike proteins proving the reusable ability during the practical applications. The samples were stored in PBS solution at room temperature for 3 days.

Modifications: On page 19, Lines 578-579 in the manuscript, we added the texts of “We did the the human tests for 4 months. All the PIDS stored in the refrigerator at -20 °C can work normally during the period of human test.”.

On page 15, Lines 436-438, we added the texts of “Furthermore, the IBS can be still active by storing it at room temperature for three days (supplementary Fig. 35) or in the refrigerator at -20 °C for several months.”.

On page 58 in the supporting information, we added supplementary Fig. 35 and related descriptions.

Reference:

1. <https://cn.sinobiological.com/antibodies/cov-spike-40150-r007>

Comment 4: The paper claimed biomarker identification using machine learning for respiratory detection. When food and drug intake is present, will the ML results be affected? Is ML capable of distinguishing features during events such as food intake? It is important to evaluate the performance of the device in terms of its ability to differentiate biomarkers from normal human activities.

Our response: We sincerely thank the reviewer for the insightful comments. We carried out the test of 21 positive cases when they are having a rest and self-isolated at home. All the participants have finished food intake and drug intake for about 2 hours before the test. The 21 negative participants have finished food intake without drug for about two hours before the test. We have added the drugs (medicine) they have taken for disease treatment in Table 4 blow. According to the information, the drug intake showed no influence on the identification ability for virus infection.

Additionally, we also tested the identification ability of the IBS module when the infected patient and healthy individual are eating food (KFC bucket meal) and drinking

water at home (see Figure R15 below). The results showed that the device still showed good ability to differentiate the exhaled virus (biomarker) from the normal human activities.

Figure R15. a Current change of the positive case by blow test. The current change ratio at 5 min was -1.1%. **b** Current change of the negative case by blow test. The current kept stable and showed no decreased tendency. All the tests are carried out when they are eating food and drinking water.

Modifications: On page 20, Lines 581-582 in the manuscript, we added the texts of “All the positive participants have completed the food intake and medicine intake for about 2 hours.”.

On page 20, Lines 589-592 in the manuscript, we added the texts of “All the negative participants have completed the food intake for about 2 hours. Additionally, the ability of PIDS to differentiate the biomarker from normal human activities (e.g., food intake) was also tested during food intake (Supplementary Fig. 44).”

On page 61 in supporting information, we added the supplementary Fig. 38 and related figure legends.

Table 4. Information of 21 positive cases

No.	Age	Gender	RR	EBT	Symptom	Vaccination	Medicine
1	34	Male	21	33.3	sore throat; sore muscle; no fever; (mild)	BNT×3	Lianhua Qingwen capsule
2	35	Male	23	33.2	sore throat; no fever; (mild)	BNT×3	Panadol
3	34	Female	17	32.9	sputum; no fever; (near asymptomatic)	BNT×3	Panadol
4	30	Male	22	33.7	sore throat; no fever; (mild)	BNT×2	Prescription medicine (expectorant, antipyretic)
5	34	Female	27	34.7	terrible sore throat; bad cough; pant; taste loss; heavy headache; nasal obstruction; high fever; (severe)	BNT×3	Prescription medicine (expectorant, antipyretic)
6	31	Female	25	34	high fever, cold, sore muscle, terrible sore throat, bad cough, heavy headache, dizzy; (severe)	BNT×3	Without medicine; Three supplements: grapefruit seed extract, ester-C, Zinc Chelate
7	55	Female	16	32.3	sputum; no fever; (near asymptomatic)	BNT×3	Panadol, Vitamin C
8	36	Male	19	32.7	runny nose; no fever; (near asymptomatic)	BNT×3	CLOFENAC SR (Diclofenac SR tablet 100 mg)
9	28	Male	26	34.5	terrible sore throat; bad cough; runny nose; dizzy; nausea; high fever; (severe)	BNT×3	Lianhua Qingwen capsule
10	42	Male	22	33.8	runny nose; cold sweat; weak; no fever; (mild)	None	Panadol
11	31	Female	23	33.5	sore throat; cough; headache; weak; no fever; (mild)	None	without medicine
12	24	Male	25	34.2	bad cough; runny nose; terrible Sore throat, dizzy, high fever; (severe)	Sinovac×2	Panadol, Cough syrup, Ryukakusan
13	21	Female	26	34.6	high Fever, terrible Sore throat; Pant; bad Cough; Runny nose; dizzy, heavy Headache; Joint pain; taste loss; (severe)	Sinovac×2	Panadol, Cough syrup, Strepsils, Ryukakusan
14	57	Male	24	34	high Fever; Sore throat; bad cough; runny nose; taste loss; (severe)	Sinovac×2	Panadol, Cough syrup, Ryukakusan
15	27	Female	26	34.5	terrible sore throat; runny nose; pant, bad cough; sputum; dizzy high fever; (severe)	BNT×3	Prescription medicine (Anti-inflammatory drug; Cold medicine)
16	23	Female	23	33.2	sore throat; cough; runny nose; no fever; (mild)	BNT×3	Panadol
17	26	Female	21	33.7	sore throat; cough; runny nose; no fever; (mild)	BNT×3	Panadol
18	26	Male	25	34.2	high fever; bad cough; runny nose; pant; terrible sore throat; (severe)	BNT×3	Panadol, Expectorant, COLTALIN
19	53	Male	18	32.6	runny nose; no fever; (near asymptomatic)	BNT×3	Cough syrup, Cold medicine
20	53	Female	17	32.3	runny nose; no fever; (near asymptomatic)	Sinovac×3	Cough syrup, Cold medicine
21	28	Female	18	32.5	itchy throat; no fever; (near	Sinovac×2,	Lianhua Qingwen capsule

Comment 5: The author claimed a varying accuracy for 42 participants. What is the overall accuracy?

Our response: Sincerely thanks for your question. For the breath test and blow test, participants could exhale more and more virus with test time. The IBS could also capture and bind with more and more virus with the test time (test duration). Therefore, the current change ratios of positive cases increased with the test time. The longer the time, the higher the accuracy. The accuracy was calculated by the formula:

$Accuracy = \frac{N_{accurate}}{N_{total}} \times 100\%$, where the $N_{accurate}$ is the accurate diagnosis number of

participants. N_{total} is the total number of participants. We took the current change ratio -0.2 % as the threshold value to differentiate positive cases from negative cases. For the blow test and breath test, we recorded and calculated the current change ratios every one minute and five minutes, respectively. Took the blow test at 1 min as an example (Figure 4i in the original manuscript), the accurate number of positive diagnoses was 10, the accurate number of negative diagnoses was 21, then the $Accuracy = \frac{10+21}{42} \times 100\% = 73.8\%$. The overall accuracy at 1 min was 73.8%. Similarly, the

overall accuracies for blow test at 2 min and 3 min were $(18+21)/42 \times 100\% = 92.9\%$ and $(18+21)/42 \times 100\% = 92.9\%$. The overall accuracies for blow test at 4 min and 5

min were all $Accuracy = \frac{21+21}{42} = 100\%$. Took the breath test at 5 min as an example

(Figure 4j in the original manuscript), the accurate number of positive diagnosis was 6, the accurate number of negative diagnosis was 21, the overall accuracy at 5 min was $(6+21)/42 \times 100\% = 64.3\%$. Similarly, the overall accuracies for breath test at 10 min, 15 min and 20 min were $(16+21)/42 \times 100\% = 88.1\%$, $(19+21)/42 \times 100\% = 95.2\%$ and $(20+21)/42 \times 100\% = 97.6\%$, respectively. The overall accuracies for breath test at 25 min and 30 min were all $(21+21)/42 \times 100\% = 100\%$.

On the other hand, this question may also refer to the accuracy for all the tests. Although such calculation is not so reasonable to evaluate the accuracy because the accuracy is related to the test time (test duration), but we also calculate the corresponding accuracy for your reference. For example, there are 10 groups in the blow test. The total test number of all positive and negative cases for blow test for 5 minutes is $21 \times 10 = 210$. The accurate numbers of positive diagnosis and negative diagnosis are $10 + 18 + 18 + 21 + 21 = 88$ and $21 + 21 + 21 + 21 + 21 = 105$, respectively. The overall accuracy can be calculated as $(88+105)/210 \times 100\% = 91.9\%$. Similarly, there are 12 groups in the breath test. The total test number of all positive and negative cases for breath test for 30 minutes is $21 \times 12 = 252$. The accurate numbers of positive diagnosis and negative diagnosis are $6 + 16 + 19 + 20 + 21 + 21 = 103$ and $21 + 21 + 21 + 21 + 21 = 126$, respectively. The overall accuracy can be calculated as $(103+126)/252 \times 100\% = 90.9\%$.

Modifications: On pages 10 and 11, Lines 312-322 in the manuscript, we added the texts of “The diagnosis accuracy was calculated using the formula, $Accuracy =$

$\frac{N_{\text{accurate}}}{N_{\text{total}}} \times 100\%$, where the N_{accurate} is the accurate diagnosis number of participants.

N_{total} is the total number of participants. Participants could exhale more and more viruses with the test time. More and more viruses could be captured by the IBS module. Therefore, the diagnosis accuracy increased with the test time. The accuracies for blow tests at different times are 73.8 % at 1 min, 92.9 % at 2 min and 3 min, 100 % at 4 min and 5 min, respectively (Fig. 4l in the manuscript). The accuracies for breath tests at different time are 64.3 % at 5 min, 88.1 % at 10 min, 95.2 % at 15 min, 97.6 % at 20 min, 100% at 25 min and 30 min, respectively. The overall accuracy of all blow tests for 5 minutes and all breath tests for 30 minutes are 91.9 % and 90.9 %, respectively (Fig. 4l).”.

Figure 4 i and j in the original manuscript. Statistical analysis of the current values by blow test and breath test, respectively. The current change ratio of -0.2 % was adopted as the threshold value to redefine the “positive” and “negative”.

Comment 6: The naïve Bayes algorithm assumes that the independence of features holds true--it is therefore needed to examine all feature correlations and their independence. However, such an assumption seems flawed in that some features selected shown in Figure 5h has high correlations, which makes applying such an algorithm questionable. Compared with parametric models such as SVM and neural network, does the naïve Bayes algorithm performs better?

Our response: Thank you sincerely for your insightful and constructive comments. We very much agree that the naïve Bayes algorithm assumes that the independence of features holds true. However, on the other hand, the Naïve Bayes assumes that each of the features contributes independently to the probability of class regardless of any possible correlations between these features. Accordingly, we assume that the correlation of different variables might not negatively influence the accuracy of classification. Furthermore, we have compared the classification accuracy by simultaneously using the two highly correlated variables (i.e., RR and EBT) in the manuscript and only using one of them (i.e., RR or EBT). The test results showed that the first method by simultaneously using RR and EBT gave the higher classification accuracy (see Figure R16a below), thus we selected these features as input. Figure R16b and Figure R16c showed the test results of removing the feature RR (Figure R16b) and EBT (Figure R16c), respectively. The corresponding accuracy is lower than that of

adding them as input features. Additionally, we also very much agree that parametric models are commonly used in many areas. Herein, we compared the accuracy using the SVM (Figure R16d) and Gaussian Naïve Bayes algorithm (Figure R16a), the results showed that the Gaussian Naïve Bayes algorithm performed better than SVM in the classification of severity.

Figure R16. Confusion matrix of the test results using different features. **a** The classification accuracy with RR and EBT was $12/13 \times 100\% \approx 92.3\%$. **b** The classification accuracy without RR was $10/13 \times 100\% \approx 76.9\%$. **c** The classification accuracy without EBT was $10/13 \times 100\% \approx 76.9\%$; **d** The classification accuracy with SVM algorithm was $7/13 \times 100\% \approx 53.8\%$.

Modifications: On Page 56 in supporting information, we added supplementary Fig. 33 and related descriptions of “Comparison of accuracy of diagnosing the viral infection without RR and EBT, respectively. Considering that RR showed high correlation with EBT ($PCC \geq 0.7$), we verified the influence of adopting both RR and EBT, and only adopting one of them. The diagnosis accuracy without RR was $(2+2+1+1+3+4)/13 \times 100\% = 100\%$. The classification accuracy without RR was $(2+1+3+4)/13 \times 100\% = 76.9\%$. The diagnosis accuracy without EBT was $(2+2+2+2+1+4)/13 \times 100\% = 100\%$. The classification accuracy without EBT was $(2+2+2+4)/13 \times 100\% = 76.9\%$. Fig. 5i in the manuscript adopted both RR and EBT. In comparison, the classification accuracy without RR or without EBT was lowered than that of adopting both RR and EBT as features. Therefore, both the RR and EBT should be adopted as features and contribute to classifying the symptom severity.”

Comment 7: The details of the ML are not given in the method. Did the author perform train-test-split for ML? Given there are 42 participants, how many data points were collected to train and evaluate the model? In Figure 5g, only 21 cases were shown but

27 variables were used. The mismatch of data size and model complexity will harm the model generalization. In Figure 5i, only 13 cases were evaluated, and this amount is inconsistent with the 21 or 42 participants. It is suggested to clarify these confusions.

Our response: Thank you sincerely for your insightful and detailed suggestions. We have performed the train-test-splitting and data shuffling on the collected dataset. The total number of volunteers is 42, and the ratio for testing set is more than 25 %, which means we have data of 29 subjects for training and 13 subjects for testing. Additionally, we feel sorry that the variables number is 28 (not 27, we corrected it in the revised manuscript) including the age, RR, EBT, vaccination, currents at different time by blowing (0 min, 1 min, 2 min, 3 min, 4 min, 5 min), currents at different time by breath (wearing mask; 0 min, 5 min, 10 min, 15 min, 20 min, 25 min, 30 min), current change ratios at different time by blowing (1 min, 2 min, 3 min, 4 min, 5min), current change ratios at different time by breath (wearing mask; 5 min, 10 min, 15 min, 20 min, 25 min, 30 min). For each of volunteer, we collected 28 features as mentioned above, but considering that most of the features showed low correlations to the positive diagnosis and contributed less to the classification, we selectively adopted 7 features as input to the model, including the age, RR, EBT, vaccination, current at 30 min by breath (wear mask), current change ratios at 5 min and 30 min by breath (wear mask).

Modifications: On page 20, Lines 598-612 in the manuscript, we added the texts of “Machine Learning. We performed the train-test-splitting and data shuffling on the collected dataset. The total number of volunteers is 42, and the ratio for testing set is more than 25 %, which means we have data of 29 subjects for training and 13 subjects for testing. For each of volunteer, we collected 28 features including the age, RR, EBT, vaccination, currents at different time by blowing (0 min, 1 min, 2 min, 3 min, 4 min, 5 min), currents at different time by breath (wearing mask; 0 min, 5 min, 10 min, 15 min, 20 min, 25 min, 30 min), current change ratios at different time by blowing (1 min, 2 min, 3 min, 4 min, 5min), current change ratios at different time by breath (wearing mask; 5 min, 10 min, 15 min, 20 min, 25 min, 30 min). Considering that most of the features showed low correlations to the positive diagnosis and contributed less to the classification, we selectively adopted 7 features as input to the model, including the age, RR, EBT, vaccination, current at 30 min by breath (wear mask), current change ratios at 5 min and 30 min by breath (wear mask).”

Comment 8: In Figure 4i, the authors defined the “positive” sample with a current ratio of -0.2% as the threshold value, however, considering the potential motion artifact, signal interference, and sensor drift, how to make sure the accuracy and long-term stability.

Our response: Thank you sincerely for your insightful and constructive comments. To ensure the accuracy and long-term stability, seven factors have been considered in our work. (1) For the structure design, we added two steel sheets (1 cm × 1cm × 1mm) on the back of IBS module and RBS module to protect them from bending and mechanical interference; (2) The IBS module, RBS module and TBS module were individually designed on different positions that can avoid the mutual interference to some extent;

(3) The front-end sensors and back-end circuit should be packaged with PDMS or other insulating materials to protect the system from short circuit in the moist aerosol environment; (4) Users can pretest the device before blow or breath diagnosis and confirm its validity to ensure the accuracy; (5) Keeping the chamber of IBS module in liquid condition and store the device at negative temperature (e.g., household refrigerator) to protect the antibody's activity and ensure the long-term stability for several months; (6) The longer the test time, the higher the accuracy. In practical applications, 4 minutes or longer is recommended for blow test. 25 minutes or longer is the recommended for breath test to ensure the near 100 % accuracy; (7) Replacing new products regularly in case of unconscious damage or destruction.

Modifications: On Page 14, Lines 426-438 in the manuscript, we added the texts of “To ensure the accuracy and long-term stability of PIDS, some measures can be considered: (1) Pretest the device before blow or breath diagnosis to confirm its validity; (2) Keep the chamber of IBS module in the liquid condition and store the device at negative temperature (e.g., household refrigerator) to protect the antibody's activity and ensure the long-term stability for several months; (3) Increase the testing time to achieve the optimum accuracy. For example, 4 minutes for blow test; 25 minutes for breath test; (4) Replace new products regularly in case of unconscious damage or destruction. The last but not the least, the IBS won't be reusable after testing the positive cases to avoid the virus transmission caused by the contaminated device. While the IBS is still reusable if the tests showed negative results. Furthermore, the IBS can be still active by storing it at room temperature for three days (supplementary Fig. 35) or in the refrigerator at $-20\text{ }^{\circ}\text{C}$ for several months.”.

Comment 9: In Figure 5d, for the temperature recording, there are several regular peaks on ambient temperature waveforms before and after breathing, please comment.

Our response: Thank you sincerely for your insightful and detailed comments. In Figure 5d in the manuscript, we showed the raw data measured and recorded by the device. The regular peaks on the ambient temperature may come from the noise signals of the device itself. To verify this point, we measured and recorded the ambient temperature again (see Figure R16a below). The ambient temperature curve showed peaks that were similar to those in Figure 5d in the manuscript. In order to calculate the breath temperature, it is necessary to filter the raw signals. An effective method is to take the average value of the signals collected over a period of time. The temperature data shown on the mobile phone was the averaged data. Considering that the noise frequency was very low ($\sim 0.04\text{ Hz}$), the breath frequency was relatively high ($\sim 0.3\text{ Hz}$), therefore, the notch filter can be used to filter out the noise signal and retain the signal of interest (see Fig. R16b below). Similarly, the noise signals of breath temperature curve disappeared after the filtration using the notch filter (see Figure R17 below).

Modifications: On Page 12, Lines 363-365 in the manuscript, we added the texts of “When record the ambient temperature, some low-frequency noise peaks ($\sim 0.04\text{ Hz}$) appeared in the curve (Fig. 5d; Supplementary Fig. 32a). These peaks could be filtered

out using the notch filter (Supplementary Fig. 32b).”.

On Page 55 in supporting information, we added supplementary Fig. 32 and related descriptions.

Figure R16. a-b The ambient temperature curves before and after filtration. **a** The ambient temperature curve was remeasured. Similar peaks with a very low frequency about 0.04 Hz were recorded, which were generated as noise signals and can be filtered using the notch filter. **b** The low-frequency noise signals disappeared after filtration using the notch filter.

Figure R17. The low-frequency noise signals in Fig. 5d in the manuscript disappeared after filtration using the notch filter.

Comment 10: Please add scale bars for all the optical images.

Our response: Thank you sincerely for your insightful and constructive suggestions. We have added the scale bars for all the optical images accordingly.

Modifications: On page 27 in the manuscript, we added the scale bars for Fig. 1e-l. On page 28 in the manuscript, we added the scale bars for Fig. 2l. On pages 17, 18, 19, 20, 21, 31 in supporting information, we added the scale bars for supplementary Fig. 2, Fig. 3, Fig. 4, Fig. 5, Fig. 6, Fig. 16, respectively.

Comments from Referee #3

Summary Comments: The manuscript titled "Wireless, battery-free, multifunctional integrated bioelectronics for respiratory pathogens monitoring and severity evaluation" by Yu et al presents a system capable of rapidly detecting SARS-CoV-2 infection through breath and blow system. The technology is showcased using various devices such as smartwatches, face masks, wristbands, and mini kits. Note that this technology has not been tested for mutated variants, but rather for the original SARS-CoV-2 strain. I think the utility of this technology is currently limited as there are already numerous tools available for monitoring SARS-CoV-2 infection at home, such as home tests based on lateral flow assays. As a result, this research fails to address the broader problem as it has focused on the wrong disease. This said, I like the extend of the technological development which I think will be of interest to the broader community. My only suggestion would be to explore detection accuracy without breath temperature and compare to the current results. I don't think it is a viable feature for classification.

Our response: We sincerely thank the reviewer's patience and time on reviewing our manuscript and gave us constructive suggestions. Accordingly, we answered all the concerns carefully point by point as follows. We sincerely hope the revisions can satisfy the reviewer's requirements.

Comment 1: Note that this technology has not been tested for mutated variants, but rather for the original SARS-CoV-2 strain.

Our response: Thank the reviewer for the comment. We had tested both the original SARS-CoV-2 strain (wild type, WT) and the typical mutated variants (e.g., Alpha, Delta, and Omicron) including the pseudo viruses and live viruses. These data were also showed in our manuscript. For example, the pseudo viruses of Alpha, Delta and Omicron were shown in Fig. 3j, Fig. 3k and Fig. 3l in the original manuscript, respectively. The live viruses of Alpha, Delta and Omicron were shown in Fig. 3n, Fig. 3o and Fig. 3p in the initial manuscript, respectively.

Figure 3. i-l Detection ability of PIDS modified with antibody 1 to pseudo viruses including WT, Alpha, Delta and Omicron. **m-p** Detection ability of PIDS modified with antibody 1 to the live viruses including WT, Alpha, Delta and Omicron. The experiment

was conducted in physical containment level 3 laboratory. PIDS showed sensitive responses to all the SARS-CoV-2 variants in the concentration range of 500 fg ml⁻¹ to 500 pg ml⁻¹.

Modifications: On page 7, line 205, we modified the subtitle as “Identification of spike proteins, pseudo virus and live virus of mutated variants”. The test results have been shown in the original Fig. 3 i-p in the manuscript.

Comment 2: I think the utility of this technology is currently limited as there are already numerous tools available for monitoring SARS-CoV-2 infection at home, such as home tests based on lateral flow assays. As a result, this research fails to address the broader problem as it has focused on the wrong disease.

Our response: Thanks for the comments. The lateral flow assays rely on the saliva and nasopharyngeal secretions by swabbed samples (i.e., nasopharyngeal and throat swab) and limited in at-home tests, which have their own limitations in large-scale screening when deal with outbreaks like COVID-19. Specifically, the technical / application limitations of current COVID-19 testing assays include:

(1) The swabs usually cause uncomfortable feelings and may not be endured by all patients, particularly infants and the elderly.

(2) The nonstandard sampling usually misses the areas with high viral loads during the swabbing that resulted in the high false-negative rate with some hurried RT-PCR tests, it even reached 58%¹.

(3) The trained staffs are required for the swab sampling to reduce the false-negative rate and avoid the inconclusive tests^{2, 3}.

(4) The samples need the time-consuming transportation for further centralized laboratory assays. The time spent for rural and remote areas could reach 48 hours or longer⁴. The results delivery of PCR needs mostly 1 or 2 days after sampling⁵.

(5) The rapid antigenic tests and sensitive molecular tests using swabbed samples have the common limitations in terms of testing procedures, which require the trained personnel and properly equipped test sites. Most of them involves challenges with the operational logistics and product supply chains for the enormous number of tests per day in every country⁵.

(6) The sampling operations have huge risks of cross infection between patients and medical workers.

Different from the existing technologies (e.g., home tests based on lateral flow assays), our study focused on digital identification of SARS-CoV-2 infection and symptom severity evaluation by gaseous aerosols in breath and blow, which enable it to detect the virus infection at an early stage and guide the patients to receive symptomatic treatment. It is not limited to the at-home tests, but also the combination with other wearable electronics in multiple-scenario applications (Supplementary Fig. 1 below). It helps to appropriately regulate medical resources in public health emergency at an early stage, predict in advance the onset of viral infections, give a window of opportunity to prevent or abate subsequent exacerbations. It addressed the broader problems and limitations far beyond the existing technologies based on lateral

flow assays.

Additionally, the existing technologies are based on the swabbed samples. From the viral transmission standpoint, respiratory activities showed a much higher priority than swabbed samples. The breath and blow (e.g., speaking, cough and sneeze) can emit thousands of SARS-CoV-2 viruses just within one minute⁶⁻⁸. The infected patients can continuously produce viral samples and emit them to environment without need of swab sampling. It suggests that the breath and blow can be alternative solutions for SARS-CoV-2 detection. In 2021, scientists published the perspective about the technical challenges, significance, and application potentials of breath analysis, which could be the most appealing approach to consistently monitor COVID-19 spread and a game changer during the pandemic⁹. Also, our test is not limited to the home test, but also the rapid detection and screening for individuals in public gathering places, for example, the subway, school, company, custom, airport, shopping mall (Supplementary Fig. 1 below).

Beyond that, we also listed the innovations and contributions of our work that advance the diagnosis of virus infection by breath and blow as follows:

- **The new method for rapid sample collection from breath and blow without manual operation.** In our work, we have created a bionic coronavirus-inspired microchannel by which we built the air-liquid interface to continuously dissolve and collect the exhaled viral droplets from breath and blow. Through this simple physical phenomenon, the air-liquid interface realized high efficiency in collecting the gaseous samples from human breath and blow within seconds that meet the requirements for rapid SARS-CoV-2 identification.
- **Onsite identification of SARS-CoV-2 by platform itself.** After collecting the breath and blow samples in the PBS solution in the chamber of microchannel, the antibodies can specifically bind with the virus spike protein and identify the antigen by the immuno biosensor of PIDS, it dispenses with the transportation for further assay using other equipment.
- **Miniaturized system integration.** In our work, we integrated three biosensing modules into an all-in-one miniaturized biosystem (size, 5cm × 5cm × 2 mm) that can be combined with various wearables and transformed into other miniaturized platforms as the handheld breathalyzer, the wearable watch/wristband/necklace, and smart patch for both breath test and blow test. The evolvable property widens the diagnostic modes and broadens its application in multiple life scenarios.
- **Multifunctional monitoring and machine learning deep analysis.** Besides collecting and identifying the exhaled SARS-CoV-2, our system can also monitor the breath rate and breath temperature. The virus signals and physical signs together with the machine learning algorithm enable us to conduct comprehensive diagnosis of virus infection and symptom severity. It may not only help the active and passive screening of populations, but also guide individuals to receive personalized therapy and lower the morbidity of severe infection.
- **Battery-free system.** We adopted the battery-free near-field communication (NFC) technique that wirelessly transmit data just by smartphone touch or walking through the NFC-enabled biosafety doors. This design avoided the frequent

replacement of the batteries, dependence of other large power sources and fixed charging positions to ensure the normal operation of the system. It also abandoned the complex connection operations of Bluetooth techniques that limited by password input between different users, especially the public gathering places.

- **Rapid diagnosis speed.** The integrated system in our work can rapidly identify the SARS-CoV-2 viruses from blow within 1 minute and breath within 5 minutes without interrupting the daily life.
- **High diagnosis accuracy.** The system in our work reached to 100% accuracy in identifying the positive and negative cases and 92% accuracy in evaluating their symptom severity.
- **Simple operation steps.** The only step the users need to do is the smartphone touch without the need of any professional training.
- **Relative mature stage.** We systematically carried out the experiment by verification of spike proteins, pseudo virus, live virus, and practical human diagnosis beyond the laboratory assay.

Moreover, COVID-19 has inflicted heavy losses on human life and global economy ever since its prevalence in December 2019. Globally, as of 28 June 2023, there have been more than 767.5 million confirmed cases of COVID-19, including 6.9 million deaths, reported to WHO¹⁰. The ongoing genome mutations and antigenic evolution increase the viral transmissibility and the risk of vaccination failure and reinfection^{11,12}, which are still life-threatening to the high-risk populations (e.g., children and the elderly with comorbidities)¹³⁻¹⁵. Additionally, more than 10 % SARS-CoV-2 infection caused the debilitating illness referring to as “long COVID” (post-acute sequelae of COVID-19).¹⁶ Furthermore, the reinfection of SARS-CoV-2 increased risks of death, hospitalization and sequelae in multiple organ systems in the acute and postacute phase.¹⁷ The rapid and early detection of infectious virus in the respiratory tract is a key step for estimating infectiousness and SARS-CoV-2 genomic surveillance,^{18,19} which help to monitor the variants of concern (VOC), contain the virus transmission and prevent the new breakout in future. All these studies showed that COVID-19 is a dangerous pandemic disease that deserves the long-term attention and research for protecting the public health security in future.

Supplementary Fig. 1 in supporting information. Outlook of diagnosis for virus infection by breath and blow using the wireless, battery-free, miniaturized, evolvable, multifunctional PIDS. **a** Compared with the conventional detections by saliva and

swabbed samples in clinical settings, breath and blow provided preferred access to respiratory pathogens. The miniaturized wearable and handhelds made pandemic diagnosis smarter and more convenient. For example, the combination with face mask, adornment (necklace), smart patch, watch, wristband, and handheld rapid detection kit. These products fully guaranteed the convenience and choosability in multiple life scenarios. **b** Wireless, battery-free, active self-diagnosis and passive screening and diagnosis in multiple life scenarios, including at-home test, outdoors, hospitals and public gathering places installed with biosafety doors. Individuals can use the evolved PIDS to conduct active self-diagnosis anywhere at any time. Medical workers can rapidly conduct passive mass screening using the smartphones, greatly saving detection time and reducing the risk of cross infection. The public gathering place (e.g., subway, school, company, custom, airport, shopping mall) can rapidly screen and diagnose the infected individuals by the wireless biosafety doors.

Modifications: The resolved broader problems, technological development, innovations, contributions on diagnosing viral infection in our work have been shown in the original supporting information. On pages 7-10 in the supporting information, we listed the “Innovations and significance by comparing our work with the pioneering works in Table 1 and Table 2”.

Comment 3: This said, I like the extend of the technological development which I think will be of interest to the broader community.

Our response: Thanks sincerely for your suggestions. Accordingly, we added another two pandemic viruses (influenza A and influenza B) to extend our technology to the broader community. The used antigen proteins involved BSA, H1N1 Hemagglutinin (HA) protein (Cat: 11055-V08H) and influenza B Hemagglutinin (HA) protein (Cat: 11053-V08H). BSA was used as the control group. IBS was modified with the influenza A H1N1 Hemagglutinin (HA) antibody (Cat: 11055-MM11) and influenza B HA antibody (Cat: 11053-R004) for specifically detecting influenza A H1N1 HA protein (Cat: 11055-V08H) and influenza B HA protein (Cat: 11053-V08H), respectively. As shown in Figure R18, IBS showed no current response to the unreactive BSA protein. IBS showed obvious current response to the H1N1 HA protein and influenza B HA protein. The current change ratios for influenza A H1N1 and influenza B were 0.87 % and 1.1%, respectively. The results indicated that our technology can be also extended to other pandemic virus detection but not limited to the SARS-CoV-2.

Figure R18. Extension of our technology to another two pandemic viruses. **a** Specific detection of influenza A H1N1. **b**. Specific detection of influenza B. The test protein concentrations were all 500 fg ml⁻¹. The test temperature was 25 °C.

Modifications: On page 57 in supporting information, we added supplementary Figure 34 and related descriptions.

On page 14, lines 418-421, we added texts of “Considering the similar transmission routes, PIDS can be also extended to other respiratory infectious diseases (e.g., influenza) by replacing the antibodies (Supplementary Fig. 34), proving the general applicability of our technology in detecting and preventing virus infection at an early stage.”.

Comment 4: My only suggestion would be to explore detection accuracy without breath temperature and compare to the current results. I don't think it is a viable feature for classification.

Our response: Sincerely thanks for your suggestions and comments. Accordingly, we explored the detection accuracy without breath temperature (i.e., EBT) (see Figure R19a below). The detection accuracy without EBT was $(2+2+2+2+1+4)/13 \times 100 \% = 100\%$. The classification accuracy of symptom severity was $(2+2+2+4)/13 \times 100 \% = 76.9 \%$. The current results were shown in Figure R19b, the detection accuracy with EBT was $(2+3+3+1+4)/13 \times 100 \% = 100 \%$. The classification accuracy of symptom severity was $(2+3+3+4)/13 \times 100 \% = 92.3 \%$. Compared to current results, the classification accuracy of symptom severity without EBT was lowered. Therefore, the breath temperature (i.e., EBT) was beneficial for classifying the symptom severity.

Additionally, the virus-induced respiratory diseases usually cause airway inflammation, followed by symptoms of fever, fatigue, sore throat, or runny nose.²⁰⁻²⁷ The measurement of exhaled breath temperature (EBT) has been suggested as a useful method to noninvasively assess the changes in the degree of airway inflammations.²⁰⁻²² The EBT rises due to the vasodilatation and increased blood flow within the airway walls.^{21,28} There are also reports about measuring the EBT and ear temperature (ET) daily for periods of between five months and two years.^{21,29} EBT rises during the viral infection (e.g., rhinovirus and influenza), affecting the respiratory system earlier than ET, providing a window of opportunity for early treatment. All these studies also proved that EBT can be a viable feature for diagnosing the virus infection and classifying the symptom severity.

Figure R19. a Detection accuracy and classification accuracy of positive and negative case without EBT. **b** Detection accuracy and classification accuracy of positive and negative case with RR and EBT.

Modifications: On page 56 in supporting information, we added supplementary Fig. 33 and related figure legends of “Comparison of accuracy of diagnosing the viral infection without RR and EBT, respectively. Considering that RR showed high correlation with EBT ($PCC \geq 0.7$), we verified the influence of adopting both RR and EBT, and only adopting one of them. The diagnosis accuracy without RR was $(2+2+1+1+3+4)/13 \times 100 \% = 100 \%$. The classification accuracy without RR was $(2+1+3+4)/13 \times 100 \% = 76.9 \%$. The diagnosis accuracy without EBT was $(2+2+2+2+1+4)/13 \times 100 \% = 100 \%$. The classification accuracy without EBT was $(2+2+2+4)/13 \times 100 \% = 76.9 \%$. Fig. 5i in the manuscript adopted both RR and EBT. In comparison, the classification accuracy without RR or without EBT was lowered than that of adopting both RR and EBT as features. Therefore, both the RR and EBT should be adopted as features and contribute to classifying the symptom severity.”.

References:

1. Pecoraro, V., Negro, A., Pirotti, T. & Trenti, T. Estimate false-negative RT-PCR rates for SARS-CoV-2. A systematic review and meta-analysis. *Eur. J. Clin. Invest.* **52**, e13706 (2022).
2. World Health Organization, Laboratory testing strategy recommendations for COVID-19, 22 March 2020 (World Health Organization, 2020).
3. Wang, W. et al. Detection of SARS-CoV-2 in different types of clinical specimens. *JAMA* **323**, 1843-1844 (2020).
4. Beeching, N. J., Fletcher, T. E. & Beadsworth, M. B. J. Covid-19: testing times. *BMJ* **369**,1403 (2020).
5. Giovannini, G., Haick, H. & Haick, H. Detecting COVID-19 from breath: A game changer for a big challenge. *ACS Sens.* **6**, 1408–1417 (2021).
6. Wang, C. C. et al. Airborne transmission of respiratory viruses. *Science* **373**, eabd9149 (2021).
7. Prather, K. A. C., Wang, C. & Schooley, R. T. Reducing transmission of SARS-CoV-2. *Science* **368**, 1422-1424 (2020).
8. Leung, N. H. L. et al. Respiratory virus shedding in exhaled breath and efficacy of face masks. *Nat. Med.* **26**, 676–680 (2020).
9. Giovannini, G., Haick, H. & Haick, H. Detecting COVID-19 from breath: A game changer for a big challenge. *ACS Sens.* **6**, 1408–1417 (2021).
10. <https://covid19.who.int/>
11. Markov, P. V., Katzourakis, A. & Stilianakis, N. I. Antigenic evolution will lead to new SARS-CoV-2 variants with unpredictable severity. *Nat. Rev. Microbiol.* **20**, 251-252 (2022).
12. Tao, K. et al. The biological and clinical significance of emerging SARS-CoV-2 variants. *Nat. Rev. Genet.* **22**, 757-773 (2021).
13. Wang, T. et al. Comorbidities and multi-organ injuries in the treatment of COVID-

19. Lancet **395**, e52 (2020).
14. Harris E. Most COVID-19 deaths worldwide were among older people. JAMA, **329**(9): 704-704 (2023).
15. Pezzullo, A. M. et al. Age-stratified infection fatality rate of COVID-19 in the non-elderly population. Environ. Res. **216**, 114655 (2023).
16. Davis H. E., et al. Long COVID: major findings, mechanisms and recommendations. Nat. Rev. Microbiol. **21**, 133-146 (2023).
17. Bowe, B. et al. Acute and postacute sequelae associated with SARS-CoV-2 reinfection. Nat. Med. **28**, 2398-2405 (2022).
18. Puhach, O. et al. SARS-CoV-2 viral load and shedding kinetics. Nat. Rev. Microbiol. **21**, 147-161 (2023).
19. Han, A. X. et al. SARS-CoV-2 diagnostic testing rates determine the sensitivity of genomic surveillance programs. Nat. Genet. **55**, 26-33 (2023).
20. Borrás, E. et al. Exhaled breath biomarkers of influenza infection and influenza vaccination. J. Breath Res. **15**, 046004 (2021).
21. Popov, T. A. et al. The added value of exhaled breath temperature in respiratory medicine. J. Breath Res. **11**, 034001 (2017).
22. Popov, T. A. et al. Measurement of exhaled breath temperature in science and clinical practice. Breathe **8**, 186-192 (2012).
23. Hassan, S. A. et al. Coronavirus (COVID-19): A Review of Clinical Features, Diagnosis, and Treatment. Cureus **12**, e7355 (2020).
24. Patterson, B. K. et al. Immune-Based Prediction of COVID-19 Severity and Chronicity Decoded Using Machine Learning. Front. Immunol. **12**, 2520 (2021).
25. Shoaib, N. et al. COVID-19 severity: Studying the clinical and demographic risk factors for adverse outcomes. PLoS One **16**, e0255999 (2021).
26. Buonsenso, D. et al. Toward a clinically based classification of disease severity for paediatric COVID-19. The Lancet. Infectious diseases **21**, 22 (2021).
27. National Health Commission & National Administration of Traditional Chinese Medicine. Diagnosis and treatment protocol for novel coronavirus pneumonia (Trial Version 7). Chinese Medical Journal **133**, 1087-1095 (2020).
28. Paredi, P. et al. Correlation of exhaled breath temperature with bronchial blood flow in asthma. Resp. Res. **6**, 1-10 (2005).
29. Popov, T. A. et al. Relationship between exhaled breath temperature and ear temperature in otherwise healthy persons during febrile infectious illness. J. Allergy Clin. Immun. **137**, AB202 (2016).

REVIEWERS' COMMENTS

Reviewer #1 (Remarks to the Author):

The authors have made extensive revisions to their manuscript to address referee inputs. I believe that the current version is now suitable for publication.

Reviewer #2 (Remarks to the Author):

The authors have fully addressed my previous comments. Now the paper is suitable for publication.

Reviewer #1

Comments summary: The authors have made extensive revisions to their manuscript to address referee inputs. I believe that the current version is now suitable for publication.

Our response: Thanks so much for the reviewer to give us positive comments and recommend the publication of our manuscript.

Reviewer #2

Comments summary: The authors have fully addressed my previous comments. Now the paper is suitable for publication.

Our response: Thanks so much for the reviewer to give us positive comments and recommend the publication of our manuscript.